# ZSPAPRUNE: ZERO-SHOT PROMPT-AWARE TOKEN PRUNING FOR VISION-LANGUAGE MODELS

## ABSTRACT

As the capabilities of Vision-Language Models (VLMs) advance, they can process increasingly large inputs, which, unlike in LLMs, generates significant visual token redundancy and leads to prohibitive inference costs. While many methods aim to reduce these costs by pruning visual tokens, existing approaches, whether based on attention or diversity, typically neglect the guidance of the text prompt and thus fail to prioritize task relevance. In this work, we propose a novel, zero-shot method that reframes the problem by introducing a prompt-aware perspective, explicitly modeling visual token pruning as a balance between task relevance and information diversity. Our hierarchical approach first selects a core set of task-relevant visual tokens and then supplements them with diversity tokens to preserve broader context. Experiments across multiple models and benchmarks show that our method achieves performance that matches or surpasses the state-of-the-art with only minimal accuracy loss, even when pruning up to 90% of the tokens. Furthermore, these gains are accompanied by significant reductions in GPU memory footprint and inference latency.

## 1 INTRODUCTION

Recent advancements in Vision-Language Models (VLMs) have led to their widespread integration into numerous real-world applications. A key mechanism distinguishing VLMs from Large Language Models (LLMs) is their reliance on vision encoders, e.g. ViT (Dosovitskiy et al., 2020) and CLIP (Radford et al., 2021), to generate sequences of visual tokens from image or video inputs. However, a fundamental challenge lies in the inherent redundancy of these tokens. Evidence from Masked Autoencoders (MAE (He et al., 2022)) that successfully reconstruct images from a small subset (25%) of visual tokens, provides a strong theoretical basis for token compression. This theoretical potential is met with a practical imperative: the growing demand for computational power is increasingly at odds with resource constraints, making efficient VLM inference a major concern. Consequently, techniques that compress visual tokens, such as pruning or merging, are becoming essential for reducing the inference costs and broadening the applicability of VLMs.

While numerous visual token pruning methods have been proposed, from early attention-based techniques like FastV (Chen et al., 2024) to recent diversity-driven approaches like DivPrune (Alvar et al., 2025), they predominantly share a critical limitation: they are prompt-agnostic. These methods operate on visual tokens in isolation, neglecting the crucial guidance provided by the text query during inference. We reframe the visual token pruning problem as one of achieving an optimal trade-off between task relevance and information diversity. As illustrated in Figure 1, an approach guided only by information diversity yields a distribution of visual tokens that is both even in its spatial spread and arbitrary in its content. Conversely, a purely task relevance-driven approach leads to an over-concentration of tokens on task-specific regions, potentially missing vital context. Achieving a deliberate balance, however, yields a far more effective and representative token subset that captures both salient information and its broader context.

To address these issues, we propose Zero-Shot Prompt-Aware Token Pruning (ZSPAPrune), a plug-and-play method designed to accelerate VLMs inference in zero-shot and resource-constrained settings. ZSPAPrune implements a hierarchical filtering strategy to select a token subset that optimally balances task relevance and information diversity. The process unfolds in three key stages: (1) the query prompt embeddings are aggregated to form a high-level semantic representation; (2) a core

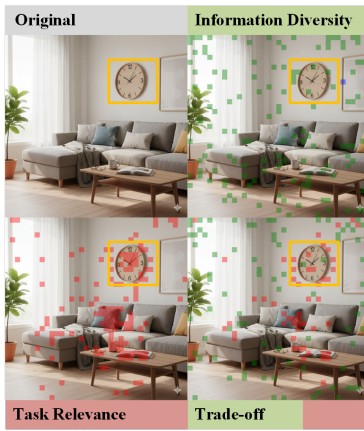
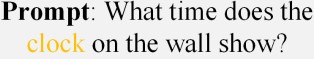
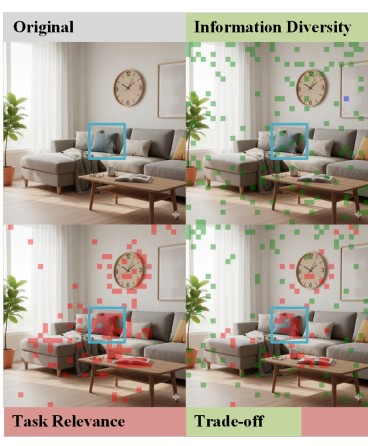
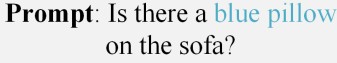
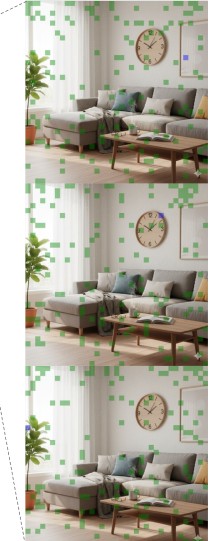

Figure 1: Trade-off between task relevance and information diversity. The two examples on the left illustrate the spatial distribution of retained tokens under three distinct settings: pure task relevance, pure information diversity, and the proposed trade-off between them. This visual comparison highlights how balancing these two metrics achieves a representative token subset, avoiding the redundancy of relevance-only methods and the potential noise of diversity-only methods. The three example images on the far right, demonstrating information diversity-based methods, illustrate their strong dependence on the initial token selection (highlighted in blue). As the initial token varies, the subsequent selection and distribution of diverse tokens (shown in green) also becomes stochastic.

set of task-relevant visual tokens is selected based on their similarity to this prompt representation; and (3) this core set is augmented with diversity tokens to ensure comprehensive visual coverage.

As a token-level pruning method, ZSPAPrune is straightforward to implement because it obviates the need to extract attention scores from the vision encoder. Furthermore, its hierarchical selection strategy decouples the core operations, making the method highly adaptable across different VLM architectures. We demonstrate its effectiveness on different models, including the early LLaVa-1.5 (Liu et al., 2024a), LLaVA-NeXT-7B (Liu et al., 2024b) and the state-of-the-art Qwen2.5-VL (Bai et al., 2025). On these models, ZSPAPrune consistently delivers significant inference acceleration while retaining approximately 90% of the original accuracy, even at an aggressive token keep rate of just 10%. Detailed results are presented in Section 4.

Our main contributions can be summarized as follows:

1. We introduce ZSPAPrune, a novel, zero-shot, plug-and-play token pruning method for VLMs. Its hierarchical and decoupled design ensures both high performance and broad applicability.

2. We reframe the visual token pruning problem as a tunable balance between task relevance and information diversity. This framework allows for adaptive adjustment of the balance based on task-specific demands, leading to a more representative subset of visual tokens.

3. We demonstrate the versatility and effectiveness of ZSPAPrune through extensive experiments. Its modular design allows for seamless integration into diverse VLM architectures, and it achieves performance that is competitive with, and in several cases surpasses, current state-of-the-art methods.

## 2 RELATED WORK

In this section, we first review recent advancements in Vision-Language Models (VLMs). We then survey and categorize existing methods for visual token pruning. Subsequently, we discuss the application of prompt-aware techniques in this domain, and conclude by highlighting the unique contributions of our proposed method.

**Vision-Language Models.** The landscape of Vision-Language Models (VLMs) is evolving rapidly, driven largely by the paradigm of integrating powerful pre-trained vision encoders with Large Language Models (LLMs). Foundational work, such as CLIP (Radford et al., 2021), demonstrated the remarkable potential of aligning image and text representations in a shared semantic space. Subsequent architectures have focused on enabling more sophisticated reasoning and generative capabilities. To bridge the modality gap, models like Flamingo (Alayrac et al., 2022) and BLIP-2 (Li et al., 2023a) introduced novel perceiver resamplers and Q-Former networks, respectively, to bridge the modality gap by transforming a variable number of visual tokens into a fixed-size set of latent queries for the LLM.

More recent and widely adopted approaches, exemplified by LLaVA (Liu et al., 2023) and MiniGPT-4 (Zhu et al., 2023), simplify this connection by using a simple linear projection layer or a two-layer MLP to map visual features from a vision encoder, like ViT-L/14 (Dosovitskiy et al., 2020), directly into the word embedding space of an LLM. This design has proven remarkably effective and has become a standard architecture. Follow-up works have continued to enhance this framework, with models like LLaVA-1.5 (Liu et al., 2024a) improving performance with better visual instruction tuning data, and recent models like Qwen-VL (Bai et al., 2023) and LLaVA-NeXT (Liu et al., 2024b) further advancing capabilities in high-resolution understanding and video reasoning. A common thread in all these models is the generation of a large sequence of visual tokens, which, while information-rich, introduces significant computational overhead during inference, motivating the need for effective token pruning.

**Visual Token Pruning.** The goal of visual token pruning is to reduce the number of visual tokens fed to the LLM, thereby decreasing latency and memory usage while minimizing performance degradation. Most existing methods are prompt-agnostic, as they operate on visual tokens without considering the input text prompt.

An early line of work leverages the internal mechanisms of the vision encoder itself. FastV (Chen et al., 2024) utilized the attention scores from the final layer of a Vision Transformer to identify and retain the most salient patches, dropping those with lower attention. While effective, this approach can be sensitive to the attention patterns of the specific vision encoder and may not capture all necessary context. Another popular strategy is to merge semantically similar tokens. ToMe (Bolya et al., 2022) introduced an efficient token merging method for ViTs by progressively combining redundant tokens based on a similarity metric rather than by purely selecting a subset.

Recognizing the importance of preserving information diversity for effective token pruning, recent methods have focused on selecting representative yet non-redundant token subsets. DivPrune (Alvar et al., 2025), for instance, formulates token pruning as a Max-Min Diversity Problem (MMDP) and solves it to obtain the desired tokens. However, the core limitation of this approach is its prompt-agnostic nature—the pruning is performed based solely on the intrinsic properties of the visual tokens.

**Prompt-Aware Pruning.** The recognition that the relevance of visual information is task-dependent has led to preliminary explorations in prompt-aware pruning. Unlike prompt-agnostic methods, these approaches aim to leverage the text prompt to guide the token selection process, focusing on the visual elements most relevant to the posed query.

Several recent studies have begun to explore this direction. For instance, PAR (Liu et al., 2024c) first applies graph-based clustering to the visual tokens and then performs a prompt-guided semantic retrieval to identify and preserve the visual token clusters most relevant to the prompt's semantics. GlimpsePrune (Zeng et al., 2025) utilizes a lightweight Visual Importance Predictor (VIP) module. This module introduces a learnable "Glimpse Token" that interacts with all visual tokens to compute a task-specific importance score for each one. Based on these scores, irrelevant visual tokens are then dynamically pruned.

These innovative methods highlight the importance of incorporating prompt-awareness. However, they are often limited in that they either require fine-tuning or fail to explicitly balance the trade-off between task relevance and information diversity. Our work, ZSPAPrune, fills this gap by proposing a zero-shot, hierarchical approach. The method first identifies a core set of highly task-relevant visual tokens guided by the prompt, then strategically supplements this core set with diversity tokens to ensure contextual completeness. This process achieves a controllable balance between relevance and diversity without the need for any retraining.

## 3 METHOD

In this section, we begin by outlining the general workflow of Vision-Language Models. We then discuss the principles of visual token pruning, and finally, provide a detailed description of our proposed ZSPAPrune method.

### 3.1 VISION-LANGUAGE MODELS

Vision-Language Models operate on an input text-image pair, $(\boldsymbol{T}, \boldsymbol{V})$. A text encoder $f_t$ and a vision encoder $f_v$ are used to process the inputs and produce sequences of text and visual token embeddings, respectively:

$$\boldsymbol{E}_t = f_t(\boldsymbol{T}), \quad \boldsymbol{E}_v = f_v(\boldsymbol{V}) \tag{1}$$

Following the initial encoding, a projection layer $p$ maps the visual tokens $\boldsymbol{E}_v$ to the text embedding space for modality alignment. These aligned tokens are then appended to the text tokens $\boldsymbol{E}_t$ and the combined sequence is fed directly into the LLM decoder $f_\phi$ to generate the output sequence $\boldsymbol{Y}$:

$$\boldsymbol{Y} = f_\phi([\boldsymbol{E}_t; p(\boldsymbol{E}_v)]) \tag{2}$$

### 3.2 VISUAL TOKEN PRUNING

The primary objective of visual token pruning is to reduce the number of redundant visual tokens, thus lowering the memory footprint and inference latency of Vision-Language Models. More formally, let $\tilde{\boldsymbol{E}}_v = p(\boldsymbol{E}_v)$ be the initial sequence of $n$ projected visual tokens. The task is to select a subsequence, $\boldsymbol{E}_{pruned}$, that forms a compact but highly representative summary of the original sequence. This selection must adhere to a predefined computational budget, such that the size of the subsequence is $|\boldsymbol{E}_{pruned}| = l$, where $l \ll n$.

### 3.3 ZSPAPRUNE

Our proposed method, ZSPAPrune, takes two sequences as input: a prompt token sequence $\boldsymbol{E}_t$ and a sequence of projected visual tokens $\tilde{\boldsymbol{E}}_v = p(\boldsymbol{E}_v)$, which are aligned with the text semantic space. These are defined as:

$$\boldsymbol{E}_t = \{t_1, t_2, t_3, ..., t_i, ..., t_m\}, \quad \tilde{\boldsymbol{E}}_v = \{v_1, v_2, v_3, ..., v_j, ..., v_n\} \tag{3}$$

Here, $\boldsymbol{E}_t$ is the sequence of $m$ prompt tokens, with each token $t_i \in \mathbb{R}^d$. Similarly, $\tilde{\boldsymbol{E}}_v$ is the sequence of $n$ visual tokens, where each token $v_j \in \mathbb{R}^d$. The term $d$ represents the shared embedding dimension. The overall process, illustrated in Figure 2, begins with an information aggregation step for the prompt tokens. This is followed by a hierarchical visual token pruning procedure, which is composed of a prompt-aware relevance selection stage and a diversity-balancing stage. The specifics of these components are detailed in Sections 3.3.1, 3.3.2, and 3.3.3, respectively.

#### 3.3.1 PROMPT SIMPLIFICATION

Calculating similarity directly against individual prompt tokens can be suboptimal. Localized details or non-essential words within the prompt can disproportionately influence the selection, overshadowing the holistic semantic intent of the query. To address this, we aim to derive a single vector that represents the high-level semantics of the entire prompt. We achieve this by applying mean pooling to the prompt token embeddings, the mathematical expression of which is:

$$\bar{t} = \frac{1}{m} \sum_{i=1}^{m} t_i \tag{4}$$

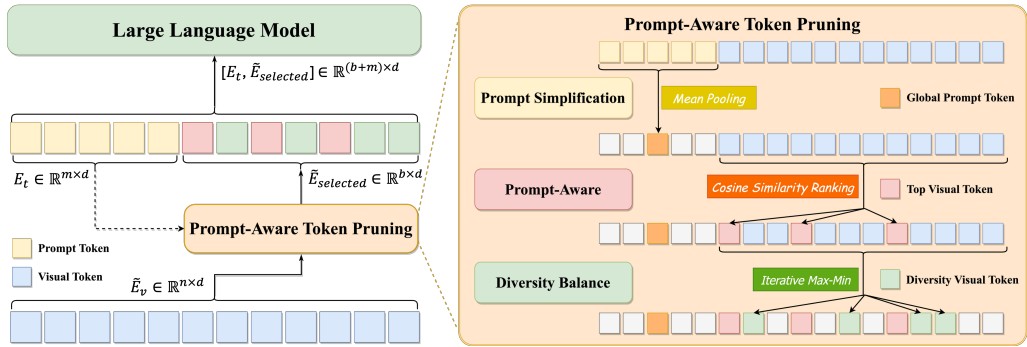

Figure 2: ZSPAPrune Architecture Framework: The left panel illustrates the inference workflow where prompt tokens ($E_t$) guide the pruning of visual tokens ($E_v$) under a pre-defined budget $b$ to produce $E_{selected}$, which is subsequently concatenated with $E_t$ for LLM input. The right panel details the specific process: (1) Prompt Simplification converts the prompt sequence into a global semantic token using mean pooling; (2) Prompt-Aware Selection retrieves top task-relevant tokens based on similarity; and (3) Diversity Balance supplements the selection with diverse tokens using an Iterative Max-Min algorithm.

where $\bar{t} \in \mathbb{R}^d$ is the resulting single vector representing the global prompt, which captures the core intent of the user's query and provides a stable guide for selecting task-relevant visual tokens.

### 3.3.2 PROMPT-AWARE

After the prompt information is aggregated, the prompt-aware module is responsible for selecting the most task-relevant visual tokens. This process consists of two main steps: **(1) calculate the task relevance between the prompt and each visual token, and (2) select the top k tokens based on these relevance scores.**

**Prompt-Visual Task Relevance Calculation.** For each visual token $v_j$ in the sequence $\tilde{E}_v = \{v_1, v_2, v_3, ..., v_j, ..., v_n\}$, its task relevance score $s_j$, is defined as the cosine similarity between the global prompt vector $\bar{t}$ and the embedding of the visual token $v_j$. This is formulated as follows:

$$s_j = \frac{\bar{t} \cdot v_j}{\|\bar{t}\|_2 \cdot \|v_j\|_2}, \quad \forall_j \in \{1, 2, ..., n\} \tag{5}$$

**Top-k Token Selection.** The relevance scores $s_j$ for all visual tokens are sorted in descending order. We then select the $k$ tokens with the highest scores. Let $\boldsymbol{I}_{core} \in \mathbb{R}^k$ be the set of indices corresponding to these top-k tokens.

$$\boldsymbol{I} = \arg \text{Top k}(s_j, k), \quad \forall_j \in \{1, 2, ..., n\} \tag{6}$$

The set of core tokens, $\tilde{\boldsymbol{E}}_{core}$, is thus defined as the subset of tokens from the original sequence $\tilde{\boldsymbol{E}}_v$ indexed by $\boldsymbol{I}_{core}$, which represents the visual tokens most relevant to the task:

$$\tilde{\boldsymbol{E}}_{core} = \{v_j | j \in \boldsymbol{I} core\} \tag{7}$$

### 3.3.3 DIVERSITY BALANCE

Once the task-relevant core tokens in $\tilde{\boldsymbol{E}}_{core}$ are selected, a diversity-enrichment stage is performed to prevent information from being over-concentrated on specific regions, which could lead to a loss of global or background context. Based on the existing set $\tilde{\boldsymbol{E}}_{core}$, this stage iteratively selects diversity tokens. At each iteration, the chosen token is the one that provides the maximum information gain, defined as the token most dissimilar to all previously selected tokens.

To supplement the core tokens, we iteratively select $\tilde{k}$ tokens from a candidate pool $\tilde{E}_{pool}$ (the original set minus the core tokens), where each selection iteration consists of the following steps:

- For each candidate token $v_x \in \tilde{E}_{pool}, \quad x \notin \boldsymbol{I}_{core}$, compute its maximum similarity to any token in the currently selected set $v_y \in \tilde{E}_{selected}, \quad y \in \boldsymbol{I}_{core}$. This value serves as a measure of the candidate token's redundancy.

- Select the candidate token $v^*$, that has the minimum redundancy score.

$$v^* = \arg\min(\max \text{ sim}(v_x, v_y)) \tag{8}$$

- Update the set:

$$\tilde{E}_{selected} \leftarrow \tilde{E}_{selected} \cup v^*, \quad \tilde{E}_{pool} \leftarrow \tilde{E}_{pool} \setminus v^* \tag{9}$$

This process is repeated for $\tilde{k}$ iterations, until the complete set of diversity tokens has been selected.Finally, the core token and the diversity token are merged to form the final pruned visual token subset $\tilde{E}_{pruned}$.This resulting subset, with a total size of budget $= k + \tilde{k}$, contains both the most task-relevant information from the core tokens and essential contextual information from the diversity tokens. In this way, it achieves a deliberate balance between task relevance and information diversity.

# 4 EXPERIMENTS

In this section, we present a comprehensive evaluation of ZSPAPrune. We conducted experiments across various models and on multiple benchmarks to assess its accuracy under different compression ratios. Furthermore, we provide a detailed efficiency analysis and present several ablation studies to validate our design choices.

## 4.1 EXPERIMENTAL SETUP

**Baselines.** We compare our method against three main baselines: (1) the Original model without any pruning; (2) DivPrune (Alvar et al., 2025), a state-of-the-art diversity-based method; and (3) an All-in Task-Relevance baseline. For a fair comparison, we reproduced all three baselines within our experimental framework. Since, to our knowledge, no existing method represents the "All-in Task-Relevance" approach, we implemented this baseline ourselves to serve as a key part.

**Models.** We selected a diverse range of Vision-Language Models (VLMs) for our evaluation: For image-based tasks, our selection included both established and recent models: LLaVA-1.5-7B (Liu et al., 2023), LLaVA-NeXT-7B (Liu et al., 2024b) and the state-of-the-art Qwen2.5-VL-7B-Instruct (Bai et al., 2025).

**Evaluation benchmarks.** We selected a comprehensive suite of benchmarks to evaluate ZSPA-Prune's performance and general applicability across a variety of tasks. For image-based tasks, we chose six diverse benchmarks: MMMU (Yue et al., 2024), GQA (Hudson & Manning, 2019), AI2D (Kembhavi et al., 2016), POPE (Li et al., 2023b), TextVQA (Singh et al., 2019), and ChartQA (Masry et al., 2022). This selection spans a wide spectrum of VQA capabilities, including yes/no questions, OCR, multiple-choice formats, and object hallucination detection, thereby demonstrating the broad utility of our method.

## 4.2 MAIN RESULTS

In this section, we evaluate our method on image benchmarks using two different model architectures: LLaVA-1.5-7B, LLaVA-NeXT-7B and the state-of-the-art Qwen2.5-VL-7B-Instruct. For all pruning methods, we set a uniform and aggressive pruning rate, removing 90% of the visual tokens. To ensure a controlled and fair comparison, all experimental parameters were held constant, with the sole exception of the relevance-diversity balance coefficient for ZSPAPrune, for which we used the optimal ratio for each specific benchmark. The primary evaluation results are presented in Table 1. This same trend is also observed on the current state-of-the-art model, Qwen2.5-VL-7B-Instruct. Specifically, the All-in Task-Relevance baseline again suffers from significantly greater accuracy loss compared to both DivPrune and ZSPAPrune.

On the LLaVA-1.5-7B model, which may have its own performance limitations, both DivPrune and ZSPAPrune show impressive results at a 90% pruning rate. For most benchmarks, the accuracy

loss is negligible, with the main exceptions being GQA (a loss of 10.7%) and TextVQA (losses of 15.9% and 16.5% for DivPrune and ZSPAPrune, respectively). Remarkably, on ChartQA, both pruning methods significantly outperform the original model, with our ZSPAPrune achieving a 22.7% improvement and DivPrune a 17.0% improvement. We attribute this surprising gain to the pruning strategy acting as a form of noise reduction; by removing redundant visual information from the charts, the method enables the model to focus on the most salient data. As a derivative version of LLaVA-1.5-7B, the LLaVA-NeXT-7B model offers enhanced overall performance. However, we found its behavior and relative trends across the different benchmarks are highly consistent with the LLaVA-1.5-7B model.

The results on the state-of-the-art Qwen2.5-VL-7B-Instruct model clearly demonstrate the advantage of ZSPAPrune's prompt-aware design. Our method outperforms DivPrune on a majority of benchmarks, with accuracy gains of 2.7% on MMMU, 1.5% on GQA, 3.8% on POPE, and 0.1% on ChartQA. This validates the importance of our approach: combining a core set of task-relevant tokens with supplementary diversity tokens significantly enhances inference accuracy by balancing relevance with diversity.

Conversely, ZSPAPrune underperforms DivPrune on AI2D and TextVQA. We attribute this to the specific nature of these benchmarks, which are less strongly task-driven. AI2D, for instance, involves complex diagrams where the task is not localized to a small region, while TextVQA is heavily reliant on dense OCR across the entire image. In such scenarios that depend more on holistic visual context, the diversity-centric approach of DivPrune has an advantage, explaining its stronger performance.

Table 1: Comparison results of ZSPAPrune and baselines on LLaVA 1.5-7B, LLaVA-NeXT-7B and Qwen2.5-VL-7B-Instruct across image-language understanding datasets. (**90% pruning rate**)

| LLaVA 1.5-7B | MMMU | GQA | AI2D | POPE | TextVQA | ChartQA |
|---|---|---|---|---|---|---|
| **Original** | 25.4 / 100.0% | 58.7 / 100.0% | 29.1 / 100.0% | 85.6 / 100.0% | 39.5 / 100.0% | 44.0 / 100.0% |
| **All-in Task-Relevance** | 25.3 / 99.6% | 43.2 / 73.6% | 28.0 / 96.2% | 71.7 / 83.8% | 11.6 / 29.4% | 50.3 / 114.3% |
| **DivPrune** | 25.4 / 100.0% | 52.4 / 89.3% | 28.7 / 98.6% | 84.7 / 98.9% | 33.2 / 84.1% | 51.5 / 117.0% |
| **ZSPAPrune** | **25.4 / 100.0%** | **52.4 / 89.3%** | 28.8 / 99.0% | 84.7 / 98.9% | 33.0 / 83.5% | **54.0 / 122.7%** |
| **LLaVA-NeXT-7B** | MMMU | GQA | AI2D | POPE | TextVQA | ChartQA |
| **Original** | 34.0 / 100.0% | 60.0 / 100.0% | 58.9 / 100.0% | 87.4 / 100.0% | 51.3 / 100.0% | 61.6 / 100.0% |
| **All-in Task-Relevance** | 31.9 / 93.8% | 41.3 / 68.8% | 56.7 / 96.3% | 33.0 / 37.8% | 9.3 / 18.1% | 54.1 / 90.2% |
| **DivPrune** | 33.8 / 99.4% | 41.0 / 68.3% | 59.1 / 100.3% | 87.0 / 99.5% | 43.7 / 85.2% | 60.5 / 98.2% |
| **ZSPAPrune** | **33.8 / 99.4%** | **41.0 / 68.3%** | 59.2 / 100.5% | 87.6 / 100.2% | 41.9 / 81.7% | **60.8 / 98.7%** |
| **Qwen2.5-VL-7B-Instruct** | MMMU | GQA | AI2D | POPE | TextVQA | ChartQA |
| **Original** | 48.2 / 100.0% | 57.7 / 100.0% | 80.6 / 100.0% | 85.8 / 100.0% | 77.9 / 100.0% | 73.8 / 100.0% |
| **All-in Task-Relevance** | 41.9 / 86.9% | 39.8 / 69.0% | 64.7 / 80.3% | 49.5 / 57.7% | 50.8 / 65.2% | 69.3 / 93.9% |
| **DivPrune** | 42.6 / 88.4% | 48.2 / 83.5% | 66.3 / 82.3% | 65.7 / 76.6% | 57.3 / 73.6% | 73.7 / 99.9% |
| **ZSPAPrune** | **43.9 / 91.1%** | **49.0 / 85.0%** | 65.3 / 81.0% | **69.0 / 80.4%** | 54.9 / 70.5% | **73.8 / 100.0%** |

An analysis of Table 1 for the LLaVA-1.5-7B model shows that both DivPrune and ZSPAPrune significantly outperform the All-in Task-Relevance baseline. This indicates that relying exclusively on prompt-guided selection, without incorporating information diversity, leads to a critical loss of information about the image itself. This lack of broader contextual relationships causes a substantial drop in performance on most benchmarks. For example, on GQA and POPE, the relevance-only baseline's accuracy is lower by 15.7% and 15.1% respectively compared to ZSPAPrune. The degradation is most severe on TextVQA, where performance plummets by approximately 54%.

## 4.3 INSIGHTS

During our experiments, we found that different benchmarks exhibit varying sensitivities to task relevance and information diversity, depending on their intrinsic characteristics. Our method, ZSPA-Prune, features a coefficient $k$ to control the ratio between core (task-relevant) and diversity tokens. A higher value prioritizes task relevance, while a lower value emphasizes information diversity. To illustrate this, we tested the accuracy of ZSPAPrune on MMMU (Yue et al., 2024), GQA (Hudson & Manning, 2019), and POPE (Li et al., 2023b) while varying $k$ from 0.1 to 0.9 (in increments of

Table 2: Comparison results of different core-diversity ratio $k$ on Qwen2.5-VL-7B-Instruct across different datasets. (**90% pruning rate**, **O** represent Original baseline)

| Qwen2.5-VL-7B-Instruct | O | 0.1 | 0.2 | 0.3 | 0.4 | 0.5 | 0.6 | 0.7 | 0.8 | 0.9 |
|---|---|---|---|---|---|---|---|---|---|---|
| **MMMU**$_{accuracy}$ | **48.2** | 42.9 | 43.3 | 43.4 | **43.9** | 43.6 | 43.1 | 43.2 | 43.0 | 41.6 |
| **GQA**$_{accuracy}$ | **57.7** | **49.0** | 48.5 | 48.2 | 47.9 | 47.0 | 46.6 | 45.5 | 44.1 | 42.8 |
| **POPE**$_{f1\text{-}score}$ | **85.8** | 68.2 | **69.0** | 65.3 | 66.4 | 65.7 | 63.7 | 61.9 | 58.0 | 53.9 |

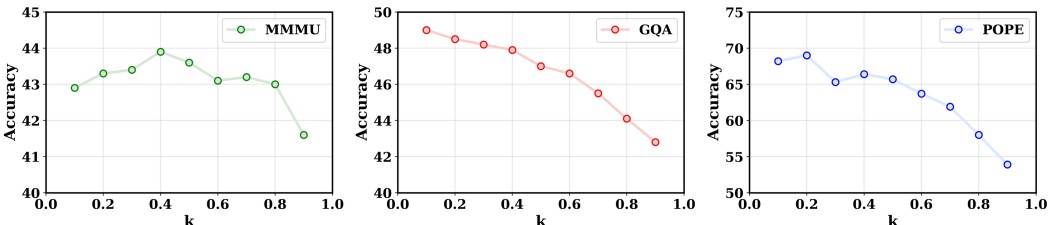

Figure 3: Comparison results of different $k$ (core-diversity ratio) on Qwen2.5-VL-7B-Instruct across MMMU, GQA and POPE

0.1). As shown in Table 2, the benchmarks display significant differences in performance across these ratios.

The accuracy trends, visualized more intuitively in Figure 3. We observe that MMMU achieves peak performance with a balancing coefficient of 0.4, indicating a need for a relatively balanced mix of tokens. In contrast, GQA and POPE perform best at coefficients of 0.1 and 0.2, respectively, which heavily favor diversity. A deeper analysis of these benchmarks reveals the underlying reasons: MMMU, with its focus on task-driven, cross-disciplinary reasoning, is highly dependent on the preservation of core, task-relevant tokens. Conversely, GQA (scene-graph reasoning) and POPE (direct yes/no judgments) rely more on a comprehensive understanding of the entire visual scene. Consequently, these benchmarks benefit from a larger proportion of diversity tokens to ensure broad semantic coverage of the image.

Based on these insights, we conclude that effective token pruning and inference acceleration for VLMs must be prompt-aware. To achieve efficient acceleration while preserving performance, pruning or merging strategies should be designed to strike an optimal balance between task relevance and information diversity, tailored to the specific characteristics of the downstream benchmark.

## 4.4 EFFICIENCY ANALYSIS

### 4.4.1 COMPLEXITY ANALYSIS

**Time Complexity.** In complete inference, given an original visual token sequence of length $N$, processing the entire sequence through an LLM with $L$ layers and embedding dimension $D$ incurs a time complexity of approximately:

$$\mathcal{O}(L \cdot (N^2 D + N D^2)) \tag{10}$$

In this equation, the $N^2 D$ term stems from the matrix multiplications within the self-attention mechanism. The $N D^2$ term arises from the linear projections and the feed-forward networks.

The inference process incorporating ZSPAPrune proceeds in two phases. Phase 1 constitutes the pruning algorithm itself, which comprises three sequential components: Prompt Simplification, Prompt-Aware Selection, and Diversity Balancing.

- **Prompt Simplification**: This step aggregates $M$ prompt tokens into a single vector representing the global semantics. It involves $(M-1)D$ additions and $D$ divisions, resulting in a time complexity of $\mathcal{O}(MD)$.

- **Prompt-Aware**: This step calculates the similarity between the aggregated prompt and $N$ visual tokens, with a complexity of $\mathcal{O}(ND)$. The subsequent top-k selection adds a complexity of $\mathcal{O}(N)$. Thus, the dominant complexity is $\mathcal{O}(ND)$.

- **Diversity Balance**: This step involves iteratively selecting $K_{\text{div}}$ diversity tokens. Each iteration requires computing similarities for the candidate tokens, incurring a cost of approximately $\mathcal{O}(ND)$. The total complexity for this step is therefore $\mathcal{O}(K_{\text{div}}ND)$.

Combining these components, the total time complexity for Phase 1 is formulated as:

$$\mathcal{O}(MD) + \mathcal{O}(ND) + \mathcal{O}(K_{\text{div}}ND) \tag{11}$$

Given that the prompt length $M$ is typically much smaller than the visual sequence length $N$ ($M \ll N$), the $\mathcal{O}(MD)$ term is negligible. Consequently, the overall complexity approximates to:

$$\mathcal{O}(MD) + \mathcal{O}((1 + K_{\text{div}})ND) \approx \mathcal{O}(K_{\text{div}}ND) \tag{12}$$

In Phase 2 (LLM Inference), the time complexity is reduced to $\mathcal{O}(L \cdot (K^2 D + KD^2))$. Here, $K$ denotes the total number of visual tokens retained, defined as $K = K_{\text{core}} + K_{\text{div}}$, where $K_{\text{core}}$ represents the core tokens relevant to the task and $K_{\text{div}}$ represents the diversity tokens.

In essence, ZSPAPrune incurs a negligible linear computational cost of $\mathcal{O}(N)$, in exchange for a substantial performance gain during the LLM inference stage by reducing the quadratic complexity from $\mathcal{O}(N^2)$ to $\mathcal{O}(K_{div}^2)$.

**Space Complexity.** Regarding space complexity, since our pruning operates directly at the token level, the space complexity of the KV Cache is reduced from $\mathcal{O}(LND)$ to $\mathcal{O}(LKD)$. Furthermore, the storage requirements for intermediate variables, such as activations, transition from scaling with $N$ (or $N^2$) to $K$ (or $K^2$). This substantial reduction in GPU memory pressure, particularly the peak memory footprint, renders our method highly suitable for deployment in resource-constrained scenarios.

### 4.4.2 EXPERIMENTAL ANALYSIS

We conducted an efficiency analysis on the LLaVA-1.5-7B model across MMMU, focusing on two key metrics: End-to-End (E2E) inference latency and the change in GPU memory consumption when our pruning strategy is applied. The results of this analysis are presented in Table 3.

Table 3: Efficiency analysis on LLaVA 1.5-7B across MMMU. (**900 samples**)

| LLaVA 1.5-7B | Original | DivPrune | ZSPAPrune |
|---|---|---|---|
| **Average E2E Latency (ms)** | 365.6 | 256.9 / ↓ **108.7** | 264.2 / ↓ **101.4** |
| **Sample 1** | | | |
| **Token (before prune)** | 576 | 576 | 576 |
| **Token (after prune)** | 576 | 58 | 58 |
| **Peak of GPU memory (MB)** | 14336.8 | 14164.8 / ↓ **172.0** | 14164.8 / ↓ **172.0** |
| **Sample 2** | | | |
| **Token (before prune)** | 576 | 576 | 576 |
| **Token (after prune)** | 576 | 58 | 58 |
| **Peak of GPU memory (MB)** | 14398.8 | 14164.8 / ↓ **234.0** | 14164.8 / ↓ **234.0** |
| **Sample 3** | | | |
| **Token (before prune)** | 576 | 576 | 576 |
| **Token (after prune)** | 576 | 58 | 58 |
| **Peak of GPU memory (MB)** | 14318.8 | 14144.8 / ↓ **174.0** | 14144.8 / ↓ **174.0** |

The analysis of Table 3 shows that ZSPAPrune significantly improves efficiency over the original baseline, reducing the average E2E inference latency by 101.4 ms. This demonstrates a substantial speedup while incurring minimal accuracy loss. When compared to DivPrune, ZSPAPrune exhibits a slightly higher latency (an increase of 7.3 ms). This overhead is an expected consequence of our

method's iterative selection process, which has a higher time complexity than DivPrune's single-pass similarity matrix calculation. Overall, we conclude that ZSPAPrune achieves a superior balance between inference speed and model performance.

To analyze GPU memory consumption, we performed a fine-grained analysis on three samples from the MMMU benchmark. With ZSPAPrune at a 90% pruning rate, the peak GPU memory usage for these three samples was reduced by 172MB, 234MB, and 174MB, respectively. The variation in savings is due to the different number of output tokens that needed to be generated for each sample. These results demonstrate a clear and substantial reduction in the overall memory footprint.

## 4.5 ABLATION STUDY

Several strategies can be used for prompt information aggregation. We compare three distinct approaches in an ablation study: (1) No Pooling: Using the initial prompt token embeddings directly, without any aggregation. (2) Max Pooling: This approach tends to focus on the subset of tokens within the prompt that have the most significant influence. (3) Mean Pooling: This approach captures a more holistic, high-level semantic representation of the prompt's overall intent. We conducted this ablation study on the MMMU and POPE benchmarks, and the results are presented in Table 4.

Table 4: Comparison results of different pooling approaches on Qwen2.5-VL-7B-Instruct across MMMU and POPE. (**90% pruning rate**, **O** represent Original baseline)

| Qwen2.5-VL-7B-Instruct | O | None | Max | Mean |
|---|---|---|---|---|
| $\mathbf{MMMU}_{\text{accuracy}}$ | **48.2** | 43.3 | 42.0 | **43.9** |
| $\mathbf{POPE}_{\text{f1-score}}$ | **85.8** | 68.2 | 67.3 | **69.0** |

The results in Table 4 indicate that the mean pooling is the most effective aggregation strategy. It consistently outperforms max pooling, achieving accuracy gains of 1.9% on MMMU and 1.7% on POPE. This suggests that our prompt-aware mechanism benefits more from a holistic, high-level representation of the prompt's overall semantics, rather than focusing only on its most salient tokens. Furthermore, compared to the no pooling baseline, mean pooling improves accuracy by 0.6% on MMMU and 0.8% on POPE, which confirms that the operation effectively integrates and extracts core semantic information from the prompt.

## 5 CONCLUSION

In this work, to combat the high inference costs of VLMs, we introduced a new paradigm for token pruning that rectifies the prompt-agnostic limitations of prior work. Our central contribution is a novel, zero-shot framework that reframes the problem as a controllable balance between task relevance and information diversity. By implementing this as a hierarchical process, which first selecting core relevant tokens, then adding diverse ones, our method achieves state-of-the-art or competitive results even at a 90% pruning rate, with significant gains in memory and speed. This study confirms that a prompt-aware, balanced pruning strategy is critical for creating efficient yet powerful VLMs, with future work potentially exploring the automated tuning of this balance.

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

## A  VISION-LANGUAGE MODEL ARCHITECTURES

### A.1  QWEN2.5-VL-INSTRUCT

Qwen2.5-VL-Instruct is a large-scale instruction-tuned vision-language model that extends the powerful Qwen2.5 language model to multimodal tasks (Bai et al., 2025). It connects a Vision Transformer (ViT) encoder to the language model backbone using a cross-attention mechanism. The model is distinguished by its ability to process high-resolution images, enabling strong performance on tasks requiring fine-grained visual understanding and optical character recognition (OCR). Its training pipeline involves large-scale pre-training followed by multi-task and instruction fine-tuning to achieve robust instruction-following and cross-modal reasoning capabilities.

### A.2  LLAVA-1.5

LLaVA-1.5 is a vision-language model that connects a pretrained CLIP visual encoder (ViT-L/14) with a large language model (Vicuna) via a simple MLP projection module (Liu et al., 2023). This architecture is efficiently trained by first pre-training the projection layer on image-text pairs, followed by end-to-end fine-tuning on a comprehensive visual instruction dataset. Compared to its predecessor, LLaVA-1.5 incorporates simple architectural improvements and an expanded, higher-quality instruction-following dataset, resulting in significantly enhanced multimodal chat and reasoning abilities.

### A.3  LLAVA-NEXT

LLaVA-NeXT is a more powerful large multimodal model that connects an advanced pretrained CLIP visual encoder (e.g., CLIP-ViT-L-336px) with a series of stronger large language models (such as Mistral-7B and Yi-34B) via an MLP projection module. The model incorporates significant architectural upgrades, notably the capability to process higher-resolution images, enabling a more detailed understanding of visual information. Its training follows an efficient two-stage paradigm: first, pre-training the projection module on image-text pairs, followed by end-to-end fine-tuning on a substantially expanded and optimized multimodal instruction dataset, which is enhanced with content from academic visual question answering (VQA), documents, and charts.

## B  EVALUATION BENCHMARKS

### B.1  EVALUATION SETTINGS

All benchmarks are locally adapted and evaluated under a unified framework. Inference outputs are assessed exclusively with discriminative metrics, without the involvement of large language models. This ensures consistent and reproducible evaluation standards across all benchmarks.

### B.2  IMAGE BENCHMARKS

#### B.2.1  MMMU

MMMU (Yue et al., 2024) is a massive multi-discipline benchmark for multimodal understanding and reasoning, designed to evaluate models on college-level problems requiring expert domain knowledge. It comprises 11.5K questions spanning six core disciplines, including Science, Art & Design, and Health & Medicine, sourced from college exams and textbooks. We employ the official test split for evaluation.

#### B.2.2  GQA

GQA (Hudson & Manning, 2019) is a benchmark for real-world visual reasoning and compositional question answering, designed to evaluate a model's ability to understand spatial relationships, object attributes, and compositional language. It consists of 113K images from Visual Genome [Krishna et al., 2017] and over 22M questions generated by a structured question engine to control for biases

and promote compositional reasoning. Following standard practice, we use the *test-dev balanced* split for evaluation.

### B.2.3 POPE

POPE (Li et al., 2023b) is a benchmark specifically designed to evaluate object hallucination in large vision-language models. It consists of binary (yes/no) questions about the presence of objects in images from the COCO validation set [Lin et al., 2014], with a balanced sampling of both present and absent objects to probe model factuality. Evaluation is based on metrics including accuracy, precision, recall, and F1 score. We report results on the official test split.

### B.3 Text-based Image Benchmarks

### B.3.1 AI2D

AI2D (Allen Institute for AI Diagrams) (Kembhavi et al., 2016) benchmark is for diagram understanding and question answering, designed to evaluate a model's ability to perform scientific and spatial reasoning. AI2D contains over 5,000 diagrams from science textbooks, each accompanied by thousands of multiple-choice questions that require parsing the diagram's layout, text, and symbols. We conduct evaluations on the official test split.

### B.3.2 TextVQA

TextVQA (Singh et al., 2019) is a visual question answering benchmark that requires reading and reasoning about text present in images. It is designed to evaluate a model's optical character recognition (OCR) and reasoning capabilities in a unified task. The dataset includes 28,408 images sourced from OpenImages [Kuznetsova et al., 2020] paired with 45,336 questions whose answers are found within the text depicted in the image. We evaluate using the validation split, consistent with prior literature.

### B.3.3 ChartQA

ChartQA (Masry et al., 2022) is a benchmark for question answering about charts, designed to test visual and logical reasoning. The task requires models to understand the structure of charts (e.g., bar, line, pie), extract relevant data, and perform numerical or logical reasoning to answer questions. It contains thousands of charts with both human-generated and machine-generated questions. Evaluations are performed on the official test split.

## C Reproducibility Statement

We have taken extensive measures to ensure the reproducibility of our results:

**Implementation Details.** Section 4.1 and 3.3 describes the experiment settings and algorithm.

**Model Modifications.** Our approach builds upon existing vision-language models with clearly documented architectural and implementation modifications. Appendix A describes them.

**Benchmarks.** We evaluate on established datasets including MMVU, GQA, POPE, AI2D, TextVQA, and ChartQA, ensuring verifiability across multiple evaluation settings. Appendix B describes them.

**Evaluation Protocol.** All experiments are conducted in a zero-shot setting under a fixed random seed, which guarantees deterministic inference. As a result, statistical significance tests or error bars are not required for interpretation.

**Open Source Release.** Upon acceptance, we will release both our algorithmic implementation and the integrated evaluation platform. The platform includes dataset-specific post-processing and evaluation scripts for all benchmarks considered, enabling direct reproduction and extension of our findings.

# D  THE USE OF LARGE LANGUAGE MODELS (LLMS)

LLMs were used only during the final editing phase of this paper to refine phrasing and correct stylistic inconsistencies. During experiments, LLMs were employed in a limited capacity for code-related tasks such as debugging, adapting model interfaces, and completing benchmark integration. Importantly, they were not involved in the design or implementation of the core algorithms. All generated content underwent rigorous human review and subsequent editing to ensure technical correctness and research integrity.

# E  ADDITIONAL EXPERIMENT

This section presents supplementary experiments conducted to provide a more comprehensive evaluation of the pruning methods under varying intensities. In addition to the aggressive setting of 10% retention rate (i.e., 90% pruning rate) discussed in the main text, we further include results for more moderate pruning settings of 30% and 50% retention rates. Table 5 provides the complete results comparing ZSPAPrune with baseline methods on LLaVA-1.5-7B, LLaVA-NeXT-7B, and Qwen2.5-VL-7B-Instruct across multiple vision-language understanding datasets. These supplementary results aim to elucidate the performance trends as the pruning intensity varies and to validate the robustness of our proposed method across different compression strengths.

Table 5: Comprehensive comparison of ZSPAPrune and baseline methods on LLaVA-1.5-7B, LLaVA-NeXT-7B, and Qwen2.5-VL-7B-Instruct across multiple vision-language understanding datasets under 10%, 30%, and 50%retention rates.

| Method | MMMU | GQA | AI2D | POPE | TextVQA | ChartQA | Average |
|---|---|---|---|---|---|---|---|
| **LLaVA 1.5-7B** | 25.4 / 100.0% | 58.7 / 100.0% | 29.1 / 100.0% | 85.6 / 100.0% | 39.5 / 100.0% | 44.0 / 100.0% | 100.0% |
| *Retention rate = 50%* | | | | | | | |
| DivPrune | 26.0 / 102.4% | 58.0 / 98.8% | 28.7 / 98.6% | 85.6 / 100.0% | 38.5 / 97.5% | 42.8 / 97.3% | 99.1% |
| ZSPAPrune | **26.0 / 102.4%** | **58.0 / 98.8%** | **28.7 / 98.6%** | **85.6 / 100.0%** | **38.5 / 97.5%** | **42.8 / 97.3%** | **99.1%** |
| *Retention rate = 30%* | | | | | | | |
| DivPrune | 26.3 / 103.5% | 57.2 / 97.4% | 28.4 / 97.6% | 85.5 / 99.9% | 37.4 / 94.7% | 45.0 / 102.3% | 99.2% |
| ZSPAPrune | **26.3 / 103.5%** | **57.2 / 97.4%** | **28.8 / 99.0%** | **85.5 / 99.9%** | **37.4 / 94.7%** | **47.7 / 108.4%** | **100.5%** |
| *Retention rate = 10%* | | | | | | | |
| All-in Task-Relevance | 25.3 / 99.6% | 43.2 / 73.6% | 28.0 / 96.2% | 71.7 / 83.8% | 11.6 / 29.4% | 50.3 / 114.3% | 82.8% |
| DivPrune | 25.4 / 100.0% | 52.4 / 89.3% | 28.7 / 98.6% | 84.7 / 98.9% | 33.2 / 84.1% | 51.5 / 117.0% | 97.9% |
| ZSPAPrune | **25.4 / 100.0%** | **52.4 / 89.3%** | **28.8 / 99.0%** | **84.7 / 98.9%** | **33.2 / 84.1%** | **54.0 / 122.7%** | **99.0%** |
| **LLaVA-NeXT-7B** | 34.0 / 100.0% | 60.0 / 100.0% | 58.9 / 100.0% | 87.4 / 100.0% | 51.3 / 100.0% | 61.6 / 100.0% | 100.0% |
| *Retention rate = 50%* | | | | | | | |
| DivPrune | 33.2 / 97.6% | 46.5 / 77.5% | 58.8 / 99.8% | 88.3 / 101.0% | 49.5 / 96.5% | 61.0 / 99.0% | 95.2% |
| ZSPAPrune | **33.2 / 97.6%** | **46.5 / 77.5%** | **58.9 / 100.0%** | **88.3 / 101.0%** | **49.5 / 96.5%** | **61.7 / 100.2%** | **95.5%** |
| *Retention rate = 30%* | | | | | | | |
| DivPrune | 33.7 / 99.1% | 44.6 / 74.3% | 59.7 / 101.4% | 88.0 / 100.7% | 47.9 / 93.4% | 60.6 / 98.4% | 94.6% |
| ZSPAPrune | **33.7 / 99.1%** | **44.6 / 74.3%** | **59.7 / 101.4%** | **88.0 / 100.7%** | **47.9 / 93.4%** | **60.8 / 98.7%** | **94.6%** |
| *Retention rate = 10%* | | | | | | | |
| All-in Task-Relevance | 31.9 / 93.8% | 41.3 / 68.8% | 56.7 / 96.3% | 33.0 / 37.8% | 9.3 / 18.1% | 54.1 / 90.2% | 67.5% |
| DivPrune | 33.8 / 99.4% | 41.0 / 68.3% | 59.1 / 100.3% | 87.0 / 99.5% | 43.7 / 85.2% | 60.5 / 98.2% | 91.8% |
| ZSPAPrune | **33.8 / 99.4%** | 41.0 / 68.3% | **59.2 / 100.5%** | **87.6 / 100.2%** | **43.7 / 85.2%** | **60.8 / 98.7%** | **92.1%** |
| **Qwen2.5-VL-7B-Instruct** | 48.2 / 100.0% | 57.7 / 100.0% | 80.6 / 100.0% | 85.8 / 100.0% | 77.9 / 100.0% | 73.8 / 100.0% | 100.0% |
| *Retention rate = 50%* | | | | | | | |
| DivPrune | 47.1 / 97.7% | 55.3 / 95.8% | 78.9 / 97.9% | 79.7 / 92.9% | 76.4 / 98.1% | 73.8 / 100.0% | 97.1% |
| ZSPAPrune | **47.1 / 97.7%** | **55.4 / 96.0%** | **78.9 / 97.9%** | **80.3 / 93.6%** | **76.4 / 98.1%** | **74.0 / 100.3%** | **97.3%** |
| *Retention rate = 30%* | | | | | | | |
| DivPrune | 44.3 / 91.9% | 53.8 / 93.2% | 75.6 / 93.8% | 77.8 / 90.7% | 72.3 / 92.8% | 73.3 / 99.3% | 93.6% |
| ZSPAPrune | **44.3 / 91.9%** | **54.0 / 93.6%** | **75.8 / 94.0%** | **77.8 / 90.7%** | **72.7 / 93.3%** | **73.9 / 100.1%** | **93.9%** |
| *Retention rate = 10%* | | | | | | | |
| All-in Task-Relevance | 41.9 / 86.9% | 39.8 / 69.0% | 64.7 / 80.3% | 49.5 / 57.7% | 50.8 / 65.2% | 69.3 / 93.9% | 75.5% |
| DivPrune | 42.6 / 88.4% | 48.2 / 83.5% | 66.3 / 82.3% | 65.7 / 76.6% | 57.3 / 73.6% | 73.7 / 99.9% | 84.1% |
| ZSPAPrune | **43.9 / 91.1%** | **49.0 / 85.0%** | **66.3 / 82.3%** | **69.0 / 80.4%** | **57.3 / 73.6%** | **73.8 / 100.0%** | **85.4%** |

We conducted a comparative analysis with VisionZip, Folder, VTW, and DivPrune across various pruning ratios, using Qwen2.5-VL-7B-Instruct as the baseline model. The results are presented in Table 6. It is important to note that unlike the other methods, VTW does not prune a fixed proportion of visual tokens; instead, it operates by evicting all visual tokens at a specific layer. For this comparison, we report the results obtained by evicting tokens at layer $k = 15$. However, we emphasize that due to this distinct mechanism, VTW lacks direct comparability with the ratio-based pruning methods.

Table 6: Comprehensive comparison of ZSPAPrune and others methods (DivPrune, Visionzip and Folder) on Qwen2.5-VL-7B-Instruct across multiple vision-language understanding datasets under 10%, 30%, and 50%retention rates.

| Method | MMMU | GQA | AI2D | POPE | TextVQA | ChartQA | Average |
|---|---|---|---|---|---|---|---|
| Qwen2.5-VL-7B-Instruct | 48.2 / 100.0% | 57.7 / 100.0% | 80.6 / 100.0% | 85.8 / 100.0% | 77.9 / 100.0% | 73.8 / 100.0% | 100.0% |
| *k = 15 , Retention rate = 50% - 60%* | | | | | | | |
| VTW | 47.2 / 97.9% | 43.9 / 76.1% | 76.1 / 94.4% | 80.1 / 93.4% | 15.8 / 20.3% | 64.4 / 87.3% | 78.2% |
| *Retention rate = 50%* | | | | | | | |
| Visionzip | 22.8 / 47.3% | 45.0 / 78.0% | 50.1 / 62.2% | 83.0 / 96.7% | 71.3 / 91.5% | 74.2 / 100.5% | 79.4% |
| Folder | 25.6 / 53.1% | 55.8 / 96.7% | 78.5 / 97.4% | 80.0 / 93.2% | 76.4 / 98.1% | 73.6 / 99.7% | 89.7% |
| DivPrune | 47.1 / 97.7% | 55.3 / 95.8% | 78.9 / 97.9% | 79.7 / 92.9% | 76.4 / 98.1% | 73.8 / 100.0% | 97.1% |
| ZSPAPrune | **47.1 / 97.7%** | 55.4 / 96.0% | **78.9 / 97.9%** | 80.3 / 93.6% | **76.4 / 98.1%** | 74.0 / 100.3% | **97.3%** |
| *Retention rate = 30%* | | | | | | | |
| Visionzip | 26.0 / 53.9% | 42.9 / 74.4% | 50.0 / 62.0% | 81.0 / 94.4% | 69.3 / 89.0% | 75.1 / 101.8% | 79.3% |
| Folder | 23.2 / 48.1% | 54.9 / 95.1% | 74.4 / 92.3% | 78.0 / 90.9% | 71.5 / 91.8% | 73.7 / 99.9% | 86.4% |
| DivPrune | 44.3 / 91.9% | 53.8 / 93.2% | 75.6 / 93.8% | 77.8 / 90.7% | 72.3 / 92.8% | 73.3 / 99.3% | 93.6% |
| ZSPAPrune | **44.3 / 91.9%** | 54.0 / 93.6% | **75.8 / 94.0%** | 77.8 / 90.7% | 72.7 / 93.3% | 73.9 / 100.1% | **93.9%** |
| *Retention rate = 10%* | | | | | | | |
| Visionzip | 28.7 / 59.5% | 37.3 / 64.6% | 47.0 / 58.3% | 73.0 / 85.1% | 52.2 / 67.0% | 74.8 / 101.4% | 72.6% |
| Folder | 22.3 / 46.3% | 51.5 / 89.3% | 65.6 / 81.4% | 69.0 / 80.4% | 51.1 / 65.6% | 73.5 / 99.6% | 77.1% |
| DivPrune | 42.6 / 88.4% | 48.2 / 83.5% | 66.3 / 82.3% | 65.7 / 76.6% | 57.3 / 73.6% | 73.7 / 99.9% | 84.1% |
| ZSPAPrune | **43.9 / 91.1%** | **49.0 / 85.0%** | **66.3 / 82.3%** | 69.0 / 80.4% | **57.3 / 73.6%** | 73.8 / 100.0% | **85.4%** |

