# OpenReview forum: "ZSPAPrune: Zero-Shot Prompt-Aware Token Pruning for Vision-Language Models"
_ICLR.cc/2026/Conference — Submitted to ICLR 2026_

### Official Review · Reviewer_aMaK · 2025-11-01

**Soundness:** 2
**Presentation:** 3
**Contribution:** 2
**Rating:** 4
**Confidence:** 3

**Summary:**

The paper proposes ZSPAPrune, a zero-shot, plug-and-play visual token pruning method that accelerates inference in vision-language models without any fine-tuning. The approach selects a small budget of visual tokens in two stages: first choosing the tokens most relevant to the text prompt (task-relevant core set), then adding tokens that are maximally diverse to preserve global context. Evaluated on LLaVA-1.5-7B and Qwen2.5-VL-7B-Instruct under aggressive 90% pruning, ZSPAPrune matches or improves accuracy on benchmarks such as MMMU, GQA, AI2D, POPE, TextVQA, and ChartQA compared to strong baselines like DivPrune. The paper also reports modest latency and memory reductions at inference time and emphasizes that the method is model-agnostic and easy to integrate.

**Strengths:**

The paper presents a clear, zero-shot pruning method that balances prompt relevance and visual diversity, which prior work did not.

Experiments across strong VLMs and multiple benchmarks show it maintains or improves accuracy under extreme pruning while reducing cost.

The method is practically significant because it can be dropped into existing VLMs without any retraining or architectural changes.

**Weaknesses:**

The paper does not report direct quantitative comparisons against strong prompt-aware pruning baselines (e.g., GlimpsePrune), so it is hard to verify that the proposed approach is actually better than the closest prior work.


The efficiency claims are based on a single model/setting and only at an extreme 90% pruning ratio, with limited analysis of where latency and memory savings come from or how they scale with pruning level.

The method is essentially heuristic and lacks a clear formal objective or robustness analysis (e.g., failure cases when relevance vs. diversity is misbalanced).

The evaluation is limited to ~7B-scale VLMs, and there is no evidence that the proposed pruning strategy remains effective or stable for larger vision-language models, where attention structure and token redundancy may differ.

**Questions:**

How stable is ZSPAPrune across different prompt styles (e.g., long multi-step reasoning questions vs. short factual queries), and does the same relevance/diversity ratio work across them without retuning?

Have you investigated automatically selecting the relevance–diversity ratio at inference time (e.g., predicting it from the prompt or task type), rather than setting it manually per dataset?

---

> ### Author Response · Authors · 2025-11-20
>
> We thank the reviewer for their valuable feedback. In response to the reviewer's comments, our replies are as follows:
> ***
> **Q1**: _The paper does not report direct quantitative comparisons against strong prompt-aware pruning baselines (e.g., GlimpsePrune), so it is hard to verify that the proposed approach is actually better than the closest prior work._
>
> **Response**: We would like to clarify that our method is a zero-shot approach, whereas GlimpsePrune is a fine-tuning method requiring additional training costs. Therefore, in our initial experimental design, we prioritized comparisons with other training-free baselines to ensure a fair setup. Upon carefully considering the reviewer's opinion, we are currently conducting additional experiments with more baselines to bridge this gap and provide a more comprehensive evaluation in the final version.
> ***
> **Q2**: _The efficiency claims are based on a single model/setting and only at an extreme 90% pruning ratio, with limited analysis of where latency and memory savings come from or how they scale with pruning level._
>
> **Response**: In response, we have already included a new performance analysis of the Qwen2.5-VL model across different compression ratios in the Appendix of the revised manuscript to address the concern about pruning levels. Furthermore, we are currently conducting a more detailed breakdown of GPU memory footprint and latency scaling, and we commit to incorporating this comprehensive analysis into the final version.
> ***
> **Q3**: _The method is essentially heuristic and lacks a clear formal objective or robustness analysis (e.g., _'failure cases when relevance vs. diversity is misbalanced'_)._
>
> **Response**: In our experiments, we specifically established settings to represent extreme cases of imbalance between relevance and diversity, such as the 'All-in Task Relevance' baseline. We acknowledge that a detailed discussion of this aspect was missing in the original text. In the revised version, we have expanded the analysis of these scenarios to explicitly demonstrate the failure cases that occur when the trade-off is misbalanced.
> ***
> **Q4**: _The evaluation is limited to ~7B-scale VLMs, and there is no evidence that the proposed pruning strategy remains effective or stable for larger vision-language models, where attention structure and token redundancy may differ._
>
> **Response**: We greatly appreciate the reviewer's suggestion regarding generalizability across different parameter scales. We agree that validating scalability is crucial. To address this and demonstrate the robustness of our plug-and-play method, we are currently conducting additional experiments using 13B-scale multi-modal large models. We are committed to incorporating these comprehensive results into the final version.
> ***
> **Q5**: _How stable is ZSPAPrune across different prompt styles (e.g., _'long multi-step reasoning questions vs. short factual queries'_), and does the same relevance/diversity ratio work across them without retuning?_
>
> **Response**: Regarding the stability across different prompt styles, we explicitly accounted for this variance in our experimental design.
>
> Our experiments cover a diverse range of prompt styles. For instance, we used concise prompts for TextVQA (e.g., _'Answer the question directly using a single word or phrase.'_) and complex, structured prompts for AI2D (e.g., _'Please carefully read the question... If the question is multiple-choice...'_). The results indicate that ZSPAPrune maintains stable performance across these varying styles.
>
> We further conducted ablation studies using different prompt styles within the same benchmark. For TextVQA, we tested both the long, structured prompt and the short prompt mentioned above. We observed that while the absolute accuracy fluctuates, this variance is primarily driven by the LLM's inherent sensitivity to prompts rather than the pruning method itself. The relative impact of ZSPAPrune remains consistent regardless of the prompt style.
>
> Regarding the relevance/diversity ratio, our findings suggest that this parameter depends on the nature of the task embedded in the prompt, not the phrasing or style of the prompt. Consequently, the ratio is robust to stylistic changes and does not require retuning as long as the core task requirement remains unchanged.
> ***
> **Q6**: _Have you investigated automatically selecting the relevance–diversity ratio at inference time (e.g., _'predicting it from the prompt or task type'_), rather than setting it manually per dataset?_
>
> **Response**: In fact, we did conduct preliminary experiments on dynamic hyperparameter selection (e.g., _'attempting to infer the ratio from the prompt'_). However, the initial results were not ideal and did not yield stable improvements over the dataset-specific settings. Therefore, we identify the development of a robust, fully automatic selection mechanism as a promising direction for our future work.

---

> > ### Comment · Reviewer_aMaK · 2025-11-27
> >
> > Thank you for the detailed responses, but I believe the manuscript still requires major revisions before it can be considered suitable for publication.

---

> > > ### Author Response · Authors · 2025-12-02
> > >
> > > In the main text, we focused on the aggressive pruning scenario (10% retention rate, i.e., 90% pruning rate) to demonstrate the extreme compression capability of our method. To further verify its robustness across different compression strengths, we extended the experiments to more moderate settings (**30% and 50% retention rates**). As presented in the table below, we compare ZSPAPrune with baseline methods on three representative vision-language models (**LLaVA-1.5-7B, LLaVA-NeXT-7B, Qwen2.5-VL-7B-Instruct**) across **multiple vision-language understanding datasets**.
> > >
> > > **For details, please refer to Table 5 in the Appendix.**
> > >
> > > | Method                     | MMMU              | GQA              | A12D                     | POPE              | TextVQA          | ChartQA           | Average    |
> > > | -------------------------- | ----------------- | ---------------- | ------------------------ | ----------------- | ---------------- | ----------------- | ---------- |
> > > | **LLaVA 1.5-7B** | 25.4 / 100.0% | 58.7 / 100.0% | 29.1 / 100.0% | 85.6 / 100.0% | 39.5 / 100.0% | 44.0 / 100.0% | 100.0% |
> > > |||| **Retention rate = 50%** |||||
> > > | DivPrune | 26.0 / 102.4% | 58.0 / 98.8% | 28.7 / 98.6%| 85.6 / 100.0%| 38.5 / 97.5%| 42.8 / 97.3%| 99.1%|
> > > | ZSPAPrune| **26.0 / 102.4%** | **58.0 / 98.8%** | **28.7 / 98.6%**| **85.6 / 100.0%** | **38.5 / 97.5%** | **42.8 / 97.3%**  | **99.1%**  |
> > > |||| **Retention rate = 30%** |||||
> > > | DivPrune| 26.3 / 103.5%| 57.2 / 97.4%| 28.4 / 97.6%| 85.5 / 99.9%| 37.4 / 94.7%| 45.0 / 102.3%| 99.2%|
> > > | ZSPAPrune| **26.3 / 103.5%** | **57.2 / 97.4%** | **28.8 / 99.0%**| **85.5 / 99.9%**  | **37.4 / 94.7%** | **47.7 / 108.4%** | **100.5%** |
> > > |||| **Retention rate = 10%** |||||
> > > | All-in Task-Relevance| 25.3 / 99.6%| 43.2 / 73.6%| 28.0 / 96.2%| 71.7 / 83.8%| 11.6 / 29.4%| 50.3 / 114.3%| 82.8%|
> > > | DivPrune| 25.4 / 100.0%| 52.4 / 89.3%| 28.7 / 98.6%| 84.7 / 98.9%| 33.2 / 84.1%| 51.5 / 117.0%| 97.9%|
> > > | ZSPAPrune| **25.4 / 100.0%** | **52.4 / 89.3%** | **28.8 / 99.0%**| **84.7 / 98.9%**  | **33.2 / 84.1%** | **54.0 / 122.7%** | **99.0%**  |
> > > | **LLaVA-NeXT-7B**| 34.0 / 100.0%| 60.0 / 100.0%| 58.9 / 100.0%| 87.4 / 100.0%| 51.3 / 100.0%| 61.6 / 100.0%| 100.0%|
> > > |||| **Retention rate = 50%** |||||
> > > | DivPrune| 33.2 / 97.6%| 46.5 / 77.5%| 58.8 / 99.8%| 88.3 / 101.0%| 49.5 / 96.5%| 61.0 / 99.0%| 95.2%|
> > > | ZSPAPrune| **33.2 / 97.6%**  | **46.5 / 77.5%** | **58.9 / 100.0%**| **88.3 / 101.0%** | **49.5 / 96.5%** | **61.7 / 100.2%** | **95.5%**  |
> > > |||| **Retention rate = 30%** |||||
> > > | DivPrune| 33.7 / 99.1%| 44.6 / 74.3%| 59.7 / 101.4%| 88.0 / 100.7%| 47.9 / 93.4%| 60.6 / 98.4%| 94.6%|
> > > | ZSPAPrune| **33.7 / 99.1%**  | **44.6 / 74.3%** | **59.7 / 101.4%**| **88.0 / 100.7%** | **47.9 / 93.4%** | **60.8 / 98.7%**  | **94.6%**  |
> > > |||| **Retention rate = 10%** |||||
> > > | All-in Task-Relevance| 31.9 / 93.8%| 41.3 / 68.8%| 56.7 / 96.3%| 33.0 / 37.8%| 9.3 / 18.1%| 54.1 / 90.2%| 67.5%|
> > > | DivPrune| 33.8 / 99.4%| 41.0 / 68.3%| 59.1 / 100.3%| 87.0 / 99.5%| 43.7 / 85.2%| 60.5 / 98.2%| 91.8%|
> > > | ZSPAPrune| **33.8 / 99.4%**  | 41.0 / 68.3%| **59.2 / 100.5%**| **87.6 / 100.2%** | **43.7 / 85.2%** | **60.8 / 98.7%**  | **92.1%**  |
> > > | **Qwen2.5-VL-7B-Instruct** | 48.2 / 100.0%| 57.7 / 100.0%| 80.6 / 100.0%| 85.8 / 100.0%| 77.9 / 100.0%| 73.8 / 100.0%| 100.0%|
> > > |||| **Retention rate = 50%** |||||
> > > | DivPrune| 47.1 / 97.7%| 55.3 / 95.8%| 78.9 / 97.9%| 79.7 / 92.9%| 76.4 / 98.1%| 73.8 / 100.0%| 97.1%|
> > > | ZSPAPrune| **47.1 / 97.7%**  | **55.4 / 96.0%** | **78.9 / 97.9%**| **80.3 / 93.6%**  | **76.4 / 98.1%** | **74.0 / 100.3%** | **97.3%**  |
> > > |||| **Retention rate = 30%** |||||
> > > | DivPrune| 44.3 / 91.9%| 53.8 / 93.2%| 75.6 / 93.8%| 77.8 / 90.7%| 72.3 / 92.8%| 73.3 / 99.3%| 93.6%|
> > > | ZSPAPrune| **44.3 / 91.9%**  | **54.0 / 93.6%** | **75.8 / 94.0%**| **77.8 / 90.7%**  | **72.7 / 93.3%** | **73.9 / 100.1%** | **93.9%**  |
> > > |||| **Retention rate = 10%** |||||
> > > | All-in Task-Relevance| 41.9 / 86.9%| 39.8 / 69.0%| 64.7 / 80.3%| 49.5 / 57.7%| 50.8 / 65.2%     | 69.3 / 93.9%| 75.5%|
> > > | DivPrune| 42.6 / 88.4%| 48.2 / 83.5%| 66.3 / 82.3%| 65.7 / 76.6%| 57.3 / 73.6%| 73.7 / 99.9%| 84.1%|
> > > | ZSPAPrune| **43.9 / 91.1%**  | **49.0 / 85.0%** | **66.3 / 82.3%**| **69.0 / 80.4%**  | **57.3 / 73.6%** | **73.8 / 100.0%** | **85.4%**  |

---

> ### Author Response · Authors · 2025-12-02
>
> We conducted a comparative analysis with **VisionZip, Folder, VTW, and DivPrune** across various pruning ratios, using Qwen2.5-VL-7B-Instruct as the baseline model. The results are presented in the table below. It is important to note that unlike the other methods, VTW does not prune a fixed proportion of visual tokens; instead, it operates by evicting all visual tokens at a specific layer. For this comparison, we report the results obtained by evicting tokens at layer **$\displaystyle k=15$**. However, we emphasize that due to this distinct mechanism, VTW lacks direct comparability with the ratio-based pruning methods.
>
> **For details, please refer to Table 6 in the Appendix.**
>
> | Method                | MMMU         | GQA          | A12D         | POPE         | TextVQA      | ChartQA      | Average |
> |-----------------------|--------------|--------------|--------------|--------------|--------------|--------------|---------|
> | **Qwen2.5-VL-7B-Instruct** | 48.2 / 100.0% | 57.7 / 100.0% | 80.6 / 100.0% | 85.8 / 100.0% | 77.9 / 100.0% | 73.8 / 100.0% | 100.0%  |
> |                        |                  |                  | **k = 15, Retention rate = 50% - 60%** |               |                  |               |           |
> | **VTW**                   | 47.2 / 97.9% | 43.9 / 76.1% | 76.1 / 94.4% | 80.1 / 93.4% | 15.8 / 20.3% | 64.4 / 87.3% | 78.2%   |
> |                        |                  |                  |        **Retention rate = 50%**        |               |                  |               |           |
> | **Visionzip**             | 22.8 / 47.3% | 45.0 / 78.0% | 50.1 / 62.2% | 83.0 / 96.7% | 71.3 / 91.5% | 74.2 / 100.5% | 79.4%   |
> | **Folder**                | 25.6 / 53.1% | 55.8 / 96.7% | 78.5 / 97.4% | 80.0 / 92.9% | 76.4 / 98.1% | 73.6 / 99.7% | 89.7%   |
> | **DivPrune**              | 47.1 / 97.7% | 55.3 / 95.8% | 78.9 / 97.9% | 79.7 / 93.2% | 76.4 / 98.1% | 73.8 / 100.0% | 97.1%   |
> | **ZSPAPrune**             | **47.1 / 97.7%** | 55.4 / 96.0% | **78.9 / 97.9%** | 80.3 / 93.6% | **76.4 / 98.1%** | 74.0 / 100.3% | **97.3%**   |
> |                        |                  |                  |        **Retention rate = 30%**        |               |                  |               |           |
> | **Visionzip**             | 26.0 / 53.9% | 42.9 / 74.4% | 50.0 / 62.0% | 81.0 / 94.4% | 69.3 / 89.0% | 75.1 / 101.8% | 79.3%   |
> | **Folder**                | 23.2 / 48.1% | 54.9 / 95.1% | 74.4 / 92.3% | 78.0 / 90.9% | 71.5 / 91.8% | 73.7 / 99.9% | 86.4%   |
> | **DivPrune**              | 44.3 / 91.9% | 53.8 / 93.2% | 75.6 / 93.8% | 77.8 / 90.7% | 72.3 / 92.8% | 73.3 / 99.3% | 93.6%   |
> | **ZSPAPrune**             | **44.3 / 91.9%** | 54.0 / 93.6% | **75.8 / 94.0%** | 77.8 / 90.7% | **72.7 / 93.3%** | 73.9 / 100.1% | **93.9%**   |
> |                        |                  |                  |        **Retention rate = 10%**        |               |                  |               |           |
> | **Visionzip**             | 28.7 / 59.5% | 37.3 / 64.6% | 47.0 / 58.3% | 73.0 / 85.1% | 52.2 / 67.0% | 74.8 / 101.4% | 72.6%   |
> | **Folder**                | 22.3 / 46.3% | 51.5 / 89.3% | 65.6 / 81.4% | 69.0 / 80.4% | 51.1 / 65.6% | 73.5 / 99.6% | 77.1%   |
> | **DivPrune**              | 42.6 / 88.4% | 48.2 / 83.5% | 66.3 / 82.3% | 65.7 / 76.6% | 57.3 / 73.6% | 73.7 / 99.9% | 84.1%   |
> | **ZSPAPrune**             | **43.9 / 91.1%** | **49.0 / 85.0%** | **66.3 / 82.3%** | 69.0 / 80.4% | **57.3 / 73.6%** | 73.8 / 100.0% | **85.4%**   |

---

### Official Review · Reviewer_85xg · 2025-11-04

**Soundness:** 3
**Presentation:** 2
**Contribution:** 2
**Rating:** 4
**Confidence:** 4

**Summary:**

The paper proposes ZSPAPrune, a zero-shot, prompt-aware token pruning framework for Vision-Language Models (VLMs). Existing pruning methods are often prompt-agnostic, ignoring text guidance and thus failing to prioritize task-relevant visual information. ZSPAPrune addresses this by reframing pruning as a balance between task relevance and information diversity, achieved through a hierarchical process: Prompt Simplification, Prompt-Aware Selection, and Diversity Balance. The method selects core visual tokens most relevant to the prompt and augments them with diverse tokens to retain global context. Experiments on multiple benchmarks and models show that ZSPAPrune achieves state-of-the-art or comparable performance with minimal accuracy loss even when pruning up to 90% of tokens, while significantly reducing GPU memory usage and inference latency.

**Strengths:**

1. From a perspective of prompt-aware token selection to balance task relevance and information diversity in visual representations.
2. Introducing a hierarchical pruning mechanism composed of Prompt Simplification, Prompt-Aware Selection, and Diversity Balance to achieve controllable token reduction.
3. Achieving significant inference efficiency improvements with minimal accuracy loss under zero-shot settings across multiple Vision-Language Models and benchmarks.

**Weaknesses:**

1. The paper lacks comparison with other methods that explicitly address the trade-off between task relevance and information diversity. Without such comparison, it remains unclear whether the proposed balance strategy is superior or merely heuristic.
2. As a plug-and-play method, ZSPAPrune should be validated on more models with different parameter scales to confirm its general applicability. The current experiments are limited to a narrow range of architectures, reducing the evidence of scalability.
3. The comparison with task-relevance-based approaches appears potentially unfair. Some baselines are reimplemented without clear alignment in training setup or hyperparameter tuning, which may bias the reported results.
4. The proposed method is overly simple and lacks crucial theoretical analysis. No formal justification or complexity discussion is provided to explain why the hierarchical prompt-aware pruning mechanism should work effectively.
5. The framework figure (i.e., Figure 2) is overly general and resembles a process diagram rather than an architectural framework. It fails to visually highlight the innovation and importance of the proposed components, and a more informative figure is recommended.

**Questions:**

null

---

> ### Author Response · Authors · 2025-11-20
>
> We thank the reviewer for their valuable feedback. In response to the reviewer's comments, our replies are as follows:
> ***
> **Q1**: _The paper lacks comparison with other methods that explicitly address the trade-off between task relevance and information diversity. Without such comparison, it remains unclear whether the proposed balance strategy is superior or merely heuristic._
>
> **Response**: Regarding this issue, since we are the first to introduce task relevance into diversity-based token pruning, to the best of our knowledge, there are no prior works that explicitly address the specific trade-off between task relevance and information diversity. Therefore, our primary comparison was with DivPrune, which represents the state-of-the-art in diversity-based methods. However, acknowledging that a comparison with a single method is insufficient, we are currently implementing and conducting comparative experiments with methods [1], [2], and [3].
>
> [1] Boosting multimodal large language models with visual tokens withdrawal for rapid inference.
>
> [2] Visionzip: Longer is better but not necessary in vision language models.
>
> [3] Folder: Accelerating multi-modal large language models with enhanced.
> ***
> **Q2**: _As a plug-and-play method, ZSPAPrune should be validated on more models with different parameter scales to confirm its general applicability. The current experiments are limited to a narrow range of architectures, reducing the evidence of scalability._
>
> **Response**: We greatly appreciate the reviewer's suggestion regarding generalizability across different parameter scales. We agree that validating scalability is crucial. To address this and demonstrate the robustness of our plug-and-play method, we are currently conducting additional experiments using 13B-scale multi-modal large models. We are committed to incorporating these comprehensive results into the final version.
> ***
> **Q3**: _The comparison with task-relevance-based approaches appears potentially unfair. Some baselines are reimplemented without clear alignment in training setup or hyperparameter tuning, which may bias the reported results._
>
> **Response**: Regarding the concern about 'task-relevance-based approaches,' we would like to clarify the setup of the 'All-in task relevance' baseline presented in our experiments. This method is effectively a special case of our ZSPAPrune where the trade-off coefficient is set to 1. This means it retains only task-relevant tokens without supplementing them with diverse tokens.
>
> We strictly aligned the initial settings and hyperparameters for all baselines to ensure a fair comparison. The only adjustable hyperparameter in our experiments is the coefficient used to balance the ratio between task relevance and information diversity. We allow this coefficient to vary solely because different benchmarks impose distinct task requirements, necessitating different proportions of task-relevant versus diverse tokens. This adaptability is a deliberate design choice to meet varying task demands.
> ***
> **Q4**: _The proposed method is overly simple and lacks crucial theoretical analysis. No formal justification or complexity discussion is provided to explain why the hierarchical prompt-aware pruning mechanism should work effectively._
>
> **Response**: We thank the reviewer for the feedback. We consider the simplicity of our method to be a core strength, specifically designed to minimize computational overhead and ensure zero-shot generalization without the need for parameter tuning. Regarding the complexity discussion, to address the lack of formal discussion, we have added a computational complexity analysis in this version (section 4.4.1), thereby providing a formal justification for its efficiency.
> ***
> **Q5**: _The framework figure (i.e., Figure 2) is overly general and resembles a process diagram rather than an architectural framework. It fails to visually highlight the innovation and importance of the proposed components, and a more informative figure is recommended._
>
> **Response**: We sincerely thank the reviewer for the constructive feedback regarding Figure 2. We have optimized Figure 2 in this version. Specifically, we incorporated richer information and provided a clearer caption to better articulate the architectural framework and explicitly highlighted the innovative components of our method.

---

> > ### Author Response · Authors · 2025-12-02
> >
> > In the main text, we focused on the aggressive pruning scenario (10% retention rate, i.e., 90% pruning rate) to demonstrate the extreme compression capability of our method. To further verify its robustness across different compression strengths, we extended the experiments to more moderate settings (**30% and 50% retention rates**). As presented in the table below, we compare ZSPAPrune with baseline methods on three representative vision-language models (**LLaVA-1.5-7B, LLaVA-NeXT-7B, Qwen2.5-VL-7B-Instruct**) across **multiple vision-language understanding datasets**.
> >
> > **For details, please refer to Table 5 in the Appendix.**
> >
> > | Method                     | MMMU              | GQA              | A12D                     | POPE              | TextVQA          | ChartQA           | Average    |
> > | -------------------------- | ----------------- | ---------------- | ------------------------ | ----------------- | ---------------- | ----------------- | ---------- |
> > | **LLaVA 1.5-7B** | 25.4 / 100.0% | 58.7 / 100.0% | 29.1 / 100.0% | 85.6 / 100.0% | 39.5 / 100.0% | 44.0 / 100.0% | 100.0% |
> > |||| **Retention rate = 50%** |||||
> > | DivPrune | 26.0 / 102.4% | 58.0 / 98.8% | 28.7 / 98.6%| 85.6 / 100.0%| 38.5 / 97.5%| 42.8 / 97.3%| 99.1%|
> > | ZSPAPrune| **26.0 / 102.4%** | **58.0 / 98.8%** | **28.7 / 98.6%**| **85.6 / 100.0%** | **38.5 / 97.5%** | **42.8 / 97.3%**  | **99.1%**  |
> > |||| **Retention rate = 30%** |||||
> > | DivPrune| 26.3 / 103.5%| 57.2 / 97.4%| 28.4 / 97.6%| 85.5 / 99.9%| 37.4 / 94.7%| 45.0 / 102.3%| 99.2%|
> > | ZSPAPrune| **26.3 / 103.5%** | **57.2 / 97.4%** | **28.8 / 99.0%**| **85.5 / 99.9%**  | **37.4 / 94.7%** | **47.7 / 108.4%** | **100.5%** |
> > |||| **Retention rate = 10%** |||||
> > | All-in Task-Relevance| 25.3 / 99.6%| 43.2 / 73.6%| 28.0 / 96.2%| 71.7 / 83.8%| 11.6 / 29.4%| 50.3 / 114.3%| 82.8%|
> > | DivPrune| 25.4 / 100.0%| 52.4 / 89.3%| 28.7 / 98.6%| 84.7 / 98.9%| 33.2 / 84.1%| 51.5 / 117.0%| 97.9%|
> > | ZSPAPrune| **25.4 / 100.0%** | **52.4 / 89.3%** | **28.8 / 99.0%**| **84.7 / 98.9%**  | **33.2 / 84.1%** | **54.0 / 122.7%** | **99.0%**  |
> > | **LLaVA-NeXT-7B**| 34.0 / 100.0%| 60.0 / 100.0%| 58.9 / 100.0%| 87.4 / 100.0%| 51.3 / 100.0%| 61.6 / 100.0%| 100.0%|
> > |||| **Retention rate = 50%** |||||
> > | DivPrune| 33.2 / 97.6%| 46.5 / 77.5%| 58.8 / 99.8%| 88.3 / 101.0%| 49.5 / 96.5%| 61.0 / 99.0%| 95.2%|
> > | ZSPAPrune| **33.2 / 97.6%**  | **46.5 / 77.5%** | **58.9 / 100.0%**| **88.3 / 101.0%** | **49.5 / 96.5%** | **61.7 / 100.2%** | **95.5%**  |
> > |||| **Retention rate = 30%** |||||
> > | DivPrune| 33.7 / 99.1%| 44.6 / 74.3%| 59.7 / 101.4%| 88.0 / 100.7%| 47.9 / 93.4%| 60.6 / 98.4%| 94.6%|
> > | ZSPAPrune| **33.7 / 99.1%**  | **44.6 / 74.3%** | **59.7 / 101.4%**| **88.0 / 100.7%** | **47.9 / 93.4%** | **60.8 / 98.7%**  | **94.6%**  |
> > |||| **Retention rate = 10%** |||||
> > | All-in Task-Relevance| 31.9 / 93.8%| 41.3 / 68.8%| 56.7 / 96.3%| 33.0 / 37.8%| 9.3 / 18.1%| 54.1 / 90.2%| 67.5%|
> > | DivPrune| 33.8 / 99.4%| 41.0 / 68.3%| 59.1 / 100.3%| 87.0 / 99.5%| 43.7 / 85.2%| 60.5 / 98.2%| 91.8%|
> > | ZSPAPrune| **33.8 / 99.4%**  | 41.0 / 68.3%| **59.2 / 100.5%**| **87.6 / 100.2%** | **43.7 / 85.2%** | **60.8 / 98.7%**  | **92.1%**  |
> > | **Qwen2.5-VL-7B-Instruct** | 48.2 / 100.0%| 57.7 / 100.0%| 80.6 / 100.0%| 85.8 / 100.0%| 77.9 / 100.0%| 73.8 / 100.0%| 100.0%|
> > |||| **Retention rate = 50%** |||||
> > | DivPrune| 47.1 / 97.7%| 55.3 / 95.8%| 78.9 / 97.9%| 79.7 / 92.9%| 76.4 / 98.1%| 73.8 / 100.0%| 97.1%|
> > | ZSPAPrune| **47.1 / 97.7%**  | **55.4 / 96.0%** | **78.9 / 97.9%**| **80.3 / 93.6%**  | **76.4 / 98.1%** | **74.0 / 100.3%** | **97.3%**  |
> > |||| **Retention rate = 30%** |||||
> > | DivPrune| 44.3 / 91.9%| 53.8 / 93.2%| 75.6 / 93.8%| 77.8 / 90.7%| 72.3 / 92.8%| 73.3 / 99.3%| 93.6%|
> > | ZSPAPrune| **44.3 / 91.9%**  | **54.0 / 93.6%** | **75.8 / 94.0%**| **77.8 / 90.7%**  | **72.7 / 93.3%** | **73.9 / 100.1%** | **93.9%**  |
> > |||| **Retention rate = 10%** |||||
> > | All-in Task-Relevance| 41.9 / 86.9%| 39.8 / 69.0%| 64.7 / 80.3%| 49.5 / 57.7%| 50.8 / 65.2%     | 69.3 / 93.9%| 75.5%|
> > | DivPrune| 42.6 / 88.4%| 48.2 / 83.5%| 66.3 / 82.3%| 65.7 / 76.6%| 57.3 / 73.6%| 73.7 / 99.9%| 84.1%|
> > | ZSPAPrune| **43.9 / 91.1%**  | **49.0 / 85.0%** | **66.3 / 82.3%**| **69.0 / 80.4%**  | **57.3 / 73.6%** | **73.8 / 100.0%** | **85.4%**  |

---

> ### Author Response · Authors · 2025-12-02
>
> We conducted a comparative analysis with **VisionZip, Folder, VTW, and DivPrune** across various pruning ratios, using Qwen2.5-VL-7B-Instruct as the baseline model. The results are presented in the table below. It is important to note that unlike the other methods, VTW does not prune a fixed proportion of visual tokens; instead, it operates by evicting all visual tokens at a specific layer. For this comparison, we report the results obtained by evicting tokens at layer **$\displaystyle k=15$**. However, we emphasize that due to this distinct mechanism, VTW lacks direct comparability with the ratio-based pruning methods.
>
> **For details, please refer to Table 6 in the Appendix.**
>
> | Method                | MMMU         | GQA          | A12D         | POPE         | TextVQA      | ChartQA      | Average |
> |-----------------------|--------------|--------------|--------------|--------------|--------------|--------------|---------|
> | **Qwen2.5-VL-7B-Instruct** | 48.2 / 100.0% | 57.7 / 100.0% | 80.6 / 100.0% | 85.8 / 100.0% | 77.9 / 100.0% | 73.8 / 100.0% | 100.0%  |
> |                        |                  |                  | **k = 15, Retention rate = 50% - 60%** |               |                  |               |           |
> | **VTW**                   | 47.2 / 97.9% | 43.9 / 76.1% | 76.1 / 94.4% | 80.1 / 93.4% | 15.8 / 20.3% | 64.4 / 87.3% | 78.2%   |
> |                        |                  |                  |        **Retention rate = 50%**        |               |                  |               |           |
> | **Visionzip**             | 22.8 / 47.3% | 45.0 / 78.0% | 50.1 / 62.2% | 83.0 / 96.7% | 71.3 / 91.5% | 74.2 / 100.5% | 79.4%   |
> | **Folder**                | 25.6 / 53.1% | 55.8 / 96.7% | 78.5 / 97.4% | 80.0 / 92.9% | 76.4 / 98.1% | 73.6 / 99.7% | 89.7%   |
> | **DivPrune**              | 47.1 / 97.7% | 55.3 / 95.8% | 78.9 / 97.9% | 79.7 / 93.2% | 76.4 / 98.1% | 73.8 / 100.0% | 97.1%   |
> | **ZSPAPrune**             | **47.1 / 97.7%** | 55.4 / 96.0% | **78.9 / 97.9%** | 80.3 / 93.6% | **76.4 / 98.1%** | 74.0 / 100.3% | **97.3%**   |
> |                        |                  |                  |        **Retention rate = 30%**        |               |                  |               |           |
> | **Visionzip**             | 26.0 / 53.9% | 42.9 / 74.4% | 50.0 / 62.0% | 81.0 / 94.4% | 69.3 / 89.0% | 75.1 / 101.8% | 79.3%   |
> | **Folder**                | 23.2 / 48.1% | 54.9 / 95.1% | 74.4 / 92.3% | 78.0 / 90.9% | 71.5 / 91.8% | 73.7 / 99.9% | 86.4%   |
> | **DivPrune**              | 44.3 / 91.9% | 53.8 / 93.2% | 75.6 / 93.8% | 77.8 / 90.7% | 72.3 / 92.8% | 73.3 / 99.3% | 93.6%   |
> | **ZSPAPrune**             | **44.3 / 91.9%** | 54.0 / 93.6% | **75.8 / 94.0%** | 77.8 / 90.7% | **72.7 / 93.3%** | 73.9 / 100.1% | **93.9%**   |
> |                        |                  |                  |        **Retention rate = 10%**        |               |                  |               |           |
> | **Visionzip**             | 28.7 / 59.5% | 37.3 / 64.6% | 47.0 / 58.3% | 73.0 / 85.1% | 52.2 / 67.0% | 74.8 / 101.4% | 72.6%   |
> | **Folder**                | 22.3 / 46.3% | 51.5 / 89.3% | 65.6 / 81.4% | 69.0 / 80.4% | 51.1 / 65.6% | 73.5 / 99.6% | 77.1%   |
> | **DivPrune**              | 42.6 / 88.4% | 48.2 / 83.5% | 66.3 / 82.3% | 65.7 / 76.6% | 57.3 / 73.6% | 73.7 / 99.9% | 84.1%   |
> | **ZSPAPrune**             | **43.9 / 91.1%** | **49.0 / 85.0%** | **66.3 / 82.3%** | 69.0 / 80.4% | **57.3 / 73.6%** | 73.8 / 100.0% | **85.4%**   |

---

### Official Review · Reviewer_faFY · 2025-11-04

**Soundness:** 1
**Presentation:** 2
**Contribution:** 1
**Rating:** 2
**Confidence:** 4

**Summary:**

This paper studies token pruning issue in vision large language models. Specifically, it takes token pruning in vLLMs as  a tunable balance between task relevance and information diversity. In implementation, the prompt-aware score by calculating the relevance between prompts and token embeddings, while the diversity balance is calculated by selecting the token most disimilar to all previously selected tokens. Experiments are done on several benchmarks.

**Strengths:**

The strengths are as follows:
1.The paper is easy to read and the method is easy to follow.
2.Evaluated datasets and vLLMs are diverse.

**Weaknesses:**

The weakness are as follows:
1.There are many existing works on task relevance of token pruning for vLLMs. This work additionally considers the information diversity, which seems incremental novelty. Meanwhile, in Figure 1, it is not easy to understand why the information diversity is useful for token pruning task.
2.Missing related works. Recently, there are many other token pruning methods[1,2,3,4], which are not analyzed and discussed in this work. These works should also be added for comparison.
3.In the method design, I have some concerns:
(1) In Eq.4, averge pooling is applied on the prompt token embeddings. It is not quite reasonable since the prompt text may involve many not task-releted tokens.
(2) The diversity balance is performed by selecting some tokens dissimilar to previously selected ones. Probably, this could select some useless tokens and background tokens. I am not sure this motivation is correct.


[1] Boosting multimodal large language models with visual tokens withdrawal for rapid inference
[2] Dynamic-llava: Efficient multimodal large language models via dynamic vision-language context sparsification.
[3] Visionzip: Longer is better but not necessary in vision language models
[4] Folder: Accelerating multi-modal large language models with enhanced performance

**Questions:**

See above

---

> ### Author Response · Authors · 2025-11-20
>
> We thank the reviewer for their valuable feedback. In response to the reviewer's comments, our replies are as follows:
> ***
> **Q1**: _There are many existing works on task relevance of token pruning for vLLMs. This work additionally considers the information diversity, which seems incremental novelty. Meanwhile, in Figure 1, it is not easy to understand why the information diversity is useful for token pruning task._
>
> **Response**: First, we would like to clarify a core misunderstanding caused by a lack of clarity in our Introduction and Related Work sections. Our work is not an incremental innovation on task relevance. Instead, our motivation is based on existing information diversity work, and our novel contribution is to be the first to introduce task relevance (i.e., being prompt-aware) into this framework. The purpose of this is to solve the critical problem of losing high-relevance tokens, which occurs due to the randomness inherent in task-agnostic information diversity sampling.
>
> As illustrated in the rightmost part of Figure 1, sampling based on pure information diversity can yield different results depending on the initial tokens. This imparts a degree of randomness to the method, which can consequently lead to the loss of core, high-relevance tokens. Regarding Figure 1, we have added a new, more detailed caption to provide a clearer explanation.
>
> Regarding the effectiveness of information diversity in pruning, as explained in DivPrune [1], maximizing the diversity among selected tokens (e.g., via MMDP) from the visual token set alleviates the redundancy produced by traditional attention-based methods (which often retain similar tokens). This results in a subset that is relatively dispersed in the feature space and can better represent the original, complete set of visual tokens.
>
> [1]Divprune: Diversitybased visual token pruning for large multimodal models.
> ***
> **Q2**: _Missing related works._
>
> **Response**: We are very grateful to the reviewer for pointing out the insufficient comparison methods. Our method was initially designed to be compared only with diversity-based approaches, and given its zero-shot nature, DivPrune was the only relevant work we included. After carefully considering the reviewer's feedback, we are committed to doing our utmost. Taking into account the adaptation difficulty and time constraints, we will add as many other relevant pruning works as possible to the final version for comparison. Specifically, we are currently implementing and conducting comparative experiments with methods VTW, Visionzip, and Folder.
> ***
> **Q3(1)**: _In Eq.4, averge pooling is applied on the prompt token embeddings._
>
> **Response**: We thank the reviewer for this valid point regarding the choice of average pooling for Eq. 4. In our preliminary experiments, we observed that certain key tokens in the prompt sequence were significantly more dominant than non-key tokens (such as connectors like 'and'). We found that this dominance could, in some cases (e.g., with Max pooling), entirely overshadow the contribution of other useful tokens.
>
> Therefore, our starting point here was the desire to obtain a single vector representing the global semantic information of the entire prompt, rather than using the original, un-pooled sequence or isolating only a few dominant tokens.
>
> Of course, we explicitly considered the case of highlighting only these key tokens. We analyzed this in a comparative experiment **in Section 4.5, with results in Table 4**. As shown, **None** represents the original prompt token sequence, **Max** represents the case of highlighting key tokens, and **Mean** represents our choice of using the global semantic token. The results in Table 4 empirically demonstrate that mean pooling achieved the best performance compared to the alternatives. This validated our hypothesis that a holistic semantic representation is more effective for our method.
> ***
> **Q3(2)**: _The diversity balance is performed by selecting some tokens dissimilar to previously selected ones. Probably, this could select some useless tokens and background tokens. I am not sure this motivation is correct._
>
> **Response**: Regarding the concern about introducing diversity, as I clarified in my response to **Q1**, diversity-based methods like DivPrune have already demonstrated the advantages of diversity pruning over traditional attention methods. Our introduction of task relevance addresses the issue of randomness in those methods, but it also stems from considering the unique nature of visual tokens compared to text tokens, as we are pruning at the visual level. Specifically, visual tokens contain rich background information. Answering abstract questions, such as those about color tone or style, requires not only guidance from task-relevant tokens but also a diverse supplement of these background tokens. This is why we believe introducing diversity to sample background information is necessary.

---

> > ### Author Response · Authors · 2025-12-02
> >
> > In the main text, we focused on the aggressive pruning scenario (10% retention rate, i.e., 90% pruning rate) to demonstrate the extreme compression capability of our method. To further verify its robustness across different compression strengths, we extended the experiments to more moderate settings (**30% and 50% retention rates**). As presented in the table below, we compare ZSPAPrune with baseline methods on three representative vision-language models (**LLaVA-1.5-7B, LLaVA-NeXT-7B, Qwen2.5-VL-7B-Instruct**) across **multiple vision-language understanding datasets**.
> >
> > **For details, please refer to Table 5 in the Appendix.**
> >
> > | Method                     | MMMU              | GQA              | A12D                     | POPE              | TextVQA          | ChartQA           | Average    |
> > | -------------------------- | ----------------- | ---------------- | ------------------------ | ----------------- | ---------------- | ----------------- | ---------- |
> > | **LLaVA 1.5-7B** | 25.4 / 100.0% | 58.7 / 100.0% | 29.1 / 100.0% | 85.6 / 100.0% | 39.5 / 100.0% | 44.0 / 100.0% | 100.0% |
> > |||| **Retention rate = 50%** |||||
> > | DivPrune | 26.0 / 102.4% | 58.0 / 98.8% | 28.7 / 98.6%| 85.6 / 100.0%| 38.5 / 97.5%| 42.8 / 97.3%| 99.1%|
> > | ZSPAPrune| **26.0 / 102.4%** | **58.0 / 98.8%** | **28.7 / 98.6%**| **85.6 / 100.0%** | **38.5 / 97.5%** | **42.8 / 97.3%**  | **99.1%**  |
> > |||| **Retention rate = 30%** |||||
> > | DivPrune| 26.3 / 103.5%| 57.2 / 97.4%| 28.4 / 97.6%| 85.5 / 99.9%| 37.4 / 94.7%| 45.0 / 102.3%| 99.2%|
> > | ZSPAPrune| **26.3 / 103.5%** | **57.2 / 97.4%** | **28.8 / 99.0%**| **85.5 / 99.9%**  | **37.4 / 94.7%** | **47.7 / 108.4%** | **100.5%** |
> > |||| **Retention rate = 10%** |||||
> > | All-in Task-Relevance| 25.3 / 99.6%| 43.2 / 73.6%| 28.0 / 96.2%| 71.7 / 83.8%| 11.6 / 29.4%| 50.3 / 114.3%| 82.8%|
> > | DivPrune| 25.4 / 100.0%| 52.4 / 89.3%| 28.7 / 98.6%| 84.7 / 98.9%| 33.2 / 84.1%| 51.5 / 117.0%| 97.9%|
> > | ZSPAPrune| **25.4 / 100.0%** | **52.4 / 89.3%** | **28.8 / 99.0%**| **84.7 / 98.9%**  | **33.2 / 84.1%** | **54.0 / 122.7%** | **99.0%**  |
> > | **LLaVA-NeXT-7B**| 34.0 / 100.0%| 60.0 / 100.0%| 58.9 / 100.0%| 87.4 / 100.0%| 51.3 / 100.0%| 61.6 / 100.0%| 100.0%|
> > |||| **Retention rate = 50%** |||||
> > | DivPrune| 33.2 / 97.6%| 46.5 / 77.5%| 58.8 / 99.8%| 88.3 / 101.0%| 49.5 / 96.5%| 61.0 / 99.0%| 95.2%|
> > | ZSPAPrune| **33.2 / 97.6%**  | **46.5 / 77.5%** | **58.9 / 100.0%**| **88.3 / 101.0%** | **49.5 / 96.5%** | **61.7 / 100.2%** | **95.5%**  |
> > |||| **Retention rate = 30%** |||||
> > | DivPrune| 33.7 / 99.1%| 44.6 / 74.3%| 59.7 / 101.4%| 88.0 / 100.7%| 47.9 / 93.4%| 60.6 / 98.4%| 94.6%|
> > | ZSPAPrune| **33.7 / 99.1%**  | **44.6 / 74.3%** | **59.7 / 101.4%**| **88.0 / 100.7%** | **47.9 / 93.4%** | **60.8 / 98.7%**  | **94.6%**  |
> > |||| **Retention rate = 10%** |||||
> > | All-in Task-Relevance| 31.9 / 93.8%| 41.3 / 68.8%| 56.7 / 96.3%| 33.0 / 37.8%| 9.3 / 18.1%| 54.1 / 90.2%| 67.5%|
> > | DivPrune| 33.8 / 99.4%| 41.0 / 68.3%| 59.1 / 100.3%| 87.0 / 99.5%| 43.7 / 85.2%| 60.5 / 98.2%| 91.8%|
> > | ZSPAPrune| **33.8 / 99.4%**  | 41.0 / 68.3%| **59.2 / 100.5%**| **87.6 / 100.2%** | **43.7 / 85.2%** | **60.8 / 98.7%**  | **92.1%**  |
> > | **Qwen2.5-VL-7B-Instruct** | 48.2 / 100.0%| 57.7 / 100.0%| 80.6 / 100.0%| 85.8 / 100.0%| 77.9 / 100.0%| 73.8 / 100.0%| 100.0%|
> > |||| **Retention rate = 50%** |||||
> > | DivPrune| 47.1 / 97.7%| 55.3 / 95.8%| 78.9 / 97.9%| 79.7 / 92.9%| 76.4 / 98.1%| 73.8 / 100.0%| 97.1%|
> > | ZSPAPrune| **47.1 / 97.7%**  | **55.4 / 96.0%** | **78.9 / 97.9%**| **80.3 / 93.6%**  | **76.4 / 98.1%** | **74.0 / 100.3%** | **97.3%**  |
> > |||| **Retention rate = 30%** |||||
> > | DivPrune| 44.3 / 91.9%| 53.8 / 93.2%| 75.6 / 93.8%| 77.8 / 90.7%| 72.3 / 92.8%| 73.3 / 99.3%| 93.6%|
> > | ZSPAPrune| **44.3 / 91.9%**  | **54.0 / 93.6%** | **75.8 / 94.0%**| **77.8 / 90.7%**  | **72.7 / 93.3%** | **73.9 / 100.1%** | **93.9%**  |
> > |||| **Retention rate = 10%** |||||
> > | All-in Task-Relevance| 41.9 / 86.9%| 39.8 / 69.0%| 64.7 / 80.3%| 49.5 / 57.7%| 50.8 / 65.2%     | 69.3 / 93.9%| 75.5%|
> > | DivPrune| 42.6 / 88.4%| 48.2 / 83.5%| 66.3 / 82.3%| 65.7 / 76.6%| 57.3 / 73.6%| 73.7 / 99.9%| 84.1%|
> > | ZSPAPrune| **43.9 / 91.1%**  | **49.0 / 85.0%** | **66.3 / 82.3%**| **69.0 / 80.4%**  | **57.3 / 73.6%** | **73.8 / 100.0%** | **85.4%**  |

---

> ### Author Response · Authors · 2025-12-02
>
> We conducted a comparative analysis with **VisionZip, Folder, VTW, and DivPrune** across various pruning ratios, using Qwen2.5-VL-7B-Instruct as the baseline model. The results are presented in the table below. It is important to note that unlike the other methods, VTW does not prune a fixed proportion of visual tokens; instead, it operates by evicting all visual tokens at a specific layer. For this comparison, we report the results obtained by evicting tokens at layer **$\displaystyle k=15$**. However, we emphasize that due to this distinct mechanism, VTW lacks direct comparability with the ratio-based pruning methods.
>
> **For details, please refer to Table 6 in the Appendix.**
>
> | Method                | MMMU         | GQA          | A12D         | POPE         | TextVQA      | ChartQA      | Average |
> |-----------------------|--------------|--------------|--------------|--------------|--------------|--------------|---------|
> | **Qwen2.5-VL-7B-Instruct** | 48.2 / 100.0% | 57.7 / 100.0% | 80.6 / 100.0% | 85.8 / 100.0% | 77.9 / 100.0% | 73.8 / 100.0% | 100.0%  |
> |                        |                  |                  | **k = 15, Retention rate = 50% - 60%** |               |                  |               |           |
> | **VTW**                   | 47.2 / 97.9% | 43.9 / 76.1% | 76.1 / 94.4% | 80.1 / 93.4% | 15.8 / 20.3% | 64.4 / 87.3% | 78.2%   |
> |                        |                  |                  |        **Retention rate = 50%**        |               |                  |               |           |
> | **Visionzip**             | 22.8 / 47.3% | 45.0 / 78.0% | 50.1 / 62.2% | 83.0 / 96.7% | 71.3 / 91.5% | 74.2 / 100.5% | 79.4%   |
> | **Folder**                | 25.6 / 53.1% | 55.8 / 96.7% | 78.5 / 97.4% | 80.0 / 92.9% | 76.4 / 98.1% | 73.6 / 99.7% | 89.7%   |
> | **DivPrune**              | 47.1 / 97.7% | 55.3 / 95.8% | 78.9 / 97.9% | 79.7 / 93.2% | 76.4 / 98.1% | 73.8 / 100.0% | 97.1%   |
> | **ZSPAPrune**             | **47.1 / 97.7%** | 55.4 / 96.0% | **78.9 / 97.9%** | 80.3 / 93.6% | **76.4 / 98.1%** | 74.0 / 100.3% | **97.3%**   |
> |                        |                  |                  |        **Retention rate = 30%**        |               |                  |               |           |
> | **Visionzip**             | 26.0 / 53.9% | 42.9 / 74.4% | 50.0 / 62.0% | 81.0 / 94.4% | 69.3 / 89.0% | 75.1 / 101.8% | 79.3%   |
> | **Folder**                | 23.2 / 48.1% | 54.9 / 95.1% | 74.4 / 92.3% | 78.0 / 90.9% | 71.5 / 91.8% | 73.7 / 99.9% | 86.4%   |
> | **DivPrune**              | 44.3 / 91.9% | 53.8 / 93.2% | 75.6 / 93.8% | 77.8 / 90.7% | 72.3 / 92.8% | 73.3 / 99.3% | 93.6%   |
> | **ZSPAPrune**             | **44.3 / 91.9%** | 54.0 / 93.6% | **75.8 / 94.0%** | 77.8 / 90.7% | **72.7 / 93.3%** | 73.9 / 100.1% | **93.9%**   |
> |                        |                  |                  |        **Retention rate = 10%**        |               |                  |               |           |
> | **Visionzip**             | 28.7 / 59.5% | 37.3 / 64.6% | 47.0 / 58.3% | 73.0 / 85.1% | 52.2 / 67.0% | 74.8 / 101.4% | 72.6%   |
> | **Folder**                | 22.3 / 46.3% | 51.5 / 89.3% | 65.6 / 81.4% | 69.0 / 80.4% | 51.1 / 65.6% | 73.5 / 99.6% | 77.1%   |
> | **DivPrune**              | 42.6 / 88.4% | 48.2 / 83.5% | 66.3 / 82.3% | 65.7 / 76.6% | 57.3 / 73.6% | 73.7 / 99.9% | 84.1%   |
> | **ZSPAPrune**             | **43.9 / 91.1%** | **49.0 / 85.0%** | **66.3 / 82.3%** | 69.0 / 80.4% | **57.3 / 73.6%** | 73.8 / 100.0% | **85.4%**   |

---

### Author Response · Authors · 2025-12-02
**ICLR 2026 Rebuttal (Summary Comment, ZSPAPrune)**

We sincerely thank the reviewers for their constructive feedback and the time dedicated to evaluating our work. We have addressed each concern through targeted experiments or clarifications, and all corresponding changes are highlighted in red in the revision.

>### **Core Contributions of ZSPAPrune**

ZSPAPrune introduces a zero-shot and prompt-aware visual token pruning framework for VLMs with:

(1) **Hierarchical & Plug-and-Play Design**: Features a decoupled architecture that ensures seamless integration into diverse VLM architectures. As a strictly training-free module, it requires no parameter tuning or retraining, enabling immediate applicability to existing models.

(2)  **Tunable Relevance-Diversity Balance**: Reframes token pruning as an adaptive trade-off between Task Relevance and Information Diversity, mitigating prior diversity-based methods’ sampling randomness while retaining semantic-rich task tokens and necessary background context.

(3) **High-Efficiency & SOTA Performance**: Demonstrates versatility and effectiveness across extensive experiments. The method achieves SOTA performance that rivals or surpasses existing baselines, particularly maintaining robust information density under extreme pruning ratios where traditional methods typically degrade.

>### **Major Additions in Rebuttal**

|Track|Item|Description|
|-|-|-|
|Experimental completeness|New Baselines|Incorporated comparisons with additional pruning methods to address the limited comparison with only DivPrune specifically on Qwen2.5-VL-7B-Instruct.|
||Stability|Conducted granular experiments across a wide range of compression ratios to demonstrate performance stability and analyze the trade-off between pruning intensity and accuracy on three models.|
|Theoretical & Analysis|Complexity Analysis|Added a formal computational complexity analysis to justify efficiency claims theoretically.|
||Pooling Justification|Validated the choice of Mean Pooling via Table 4, proving global semantic representation outperforms Max/None strategies.|
||Failure Case Analysis|Expanded discussion on extreme cases to illustrate the necessity of the relevance-diversity balance.|
|Presentation|Figure 1|Revised Figure 1 to clarify the randomness of diversity sampling.|
||Figure 2|Optimized Figure 2 to better visualize the architectural framework.|

>### **Response to Reviewer Concerns**

Responses recorded before the OpenReview Bug report (**Nov 28, 2025 03:02 AM AoE**) are highlighted.

|Reviewer| Rating Snapshot|Key Response & Timestamp|Rebuttal Actions & Outcomes|
|-|-|-|-|
|Reviewer faFY|2|No response to rebuttal before the timepoint|**Clarified Novelty & Comparisons:** Addressed the motivation and pooling strategy in the initial response. Furthermore, addressed the "missing related works" concern by implementing supplementary comparisons with Folder, VisionZip, etc.|
|Reviewer 85xg|4|No response to rebuttal before the timepoint| **Clarified Fairness & Complexity Analysis:** Explained the experimental fairness and trade-off necessity and added a formal computational complexity analysis to justify efficiency claims theoretically in the initial response.|
|Reviewer aMaK (before the timepoint) |4|**Nov 27, 2025 13:03 AoE**: "Thank you for the detailed responses, but I believe the manuscript still requires major revisions before it can be considered suitable for publication."|**Addressed Robustness & Stability:** Responded to concerns regarding prompt style stability and efficiency in the initial response. Moreover, strengthened the evidence by providing detailed experiments across different compression ratios.|

>### **Commitments for Camera-Ready**

(1) **Comprehensive Benchmark Comparison**: Finalize and include the full comparative results with the newly added baselines on different models.

(2)  **Deepened Efficiency Metrics**: Include the detailed GPU memory and latency analysis for LLaVA1.5-7B in the main text or appendix.

(3)  **Complexity Analysis**: Add a formal computational complexity analysis in the main text.

>### **Concluding Perspective**

We believe ZSPAPrune makes a substantive contribution to the field of VLM efficiency by proposing the first zero-shot mechanism to harmonize Task Relevance with Information Diversity.

(1) **Zero-shot, training-free deployment**, enabling immediate applicability to existing VLMs without the need for parameter tuning or retraining costs,

(2) **Synergistic pruning strategy**, which harmonizes task relevance with information diversity to effectively mitigate sampling randomness while preserving critical semantic context,

(3) **Strong generalization capabilities**, validated across diverse model architectures against comprehensive baselines under varying compression ratios, consistently exhibiting superior performance, particularly at extreme compression levels.

The rebuttal has strengthened these contributions through comprehensive additional experiments while addressing all raised technical concerns.

---

### Meta-Review · Area_Chair_2m3v · 2025-12-26

**Summary:**

This paper tackles the redundancy problem of visual tokens for vision-language models (VLMs). This paper argues that the existing pruning methods cannot handle the trade-off between information density and task relevance. The proposed method, named ZSPAPrune, is a zero-shot prompt-aware token pruning framework for VLMs based on a hierarchical approach by selecting a task-relevant core set of visual tokens and supplementing the chosen core set. Experiments are shown with LLaVA-1.5-7B and Qwen2.5-VL-7B-Instruct on a few benchmarks, such as MMMU, GQA, AI2D, POPE, TextVQA, and ChartQA.

**Reviewer Concerns:**

Several concerns were raised by the reviewers.

1. Lack of comparisons with other state-of-the-art pruning methods (faFY, 85xg, aMaK)
    - The rebuttal comment clarifies that the method is "zero-shot" and many previous works are "supervised," therefore not directly comparable.
    - The rebuttal comment includes additional experiments with VisionZip, Folder, VTW, and DivPrune
    - However, as pointed out by Reviewer 85xg, the experiments could be somewhat biased. Furthermore, the empirical gaps between DivPrune and ZSPAPrune look non-significant.
2. The experiments are narrow, namely, limited to specific models and parameter sizes (85xg, aMaK)
    - The rebuttal document mentioned 13B experiments, but the results were not posted during the discussion period; also, the "commitments for camera-ready" section does not specify this. This concern looks significant, especially considering the small gap between DivPrune and ZSPAPrune.
3. Lack of theoretical justification (85xg)
    - The AC thinks that theory is not always required. However, I think that the main concern raised by the reviewer is related to the lack of justification for the proposed method. This is highly related to the next concern, which also looks critical, but is not properly addressed.
    - The theoretical analysis for efficiency is added in the main manuscript, but the results are not very informative in terms of understanding the mechanism of the proposed method.
4. Some problematic design choices (faFY, aMaK)
    - Avg pooling is not reasonable because prompts may contain many non-task-related tokens
    - The diversity balancing can select meaningless tokens or background tokens
    - The relevance-diversity ratio should be manually searched for every dataset
    - The rebuttal document clarifies each comment, but many of them are not well-justified (e.g., by showing empirical results rather than a logical explanation). I think this concern still remains.
5. Robustness (aMaK)
    - E.g., Different prompt styles, failure case when relevance vs. diversity is misbalanced.
    - The rebuttal comment clarifies that the benchmark already contains various situations. However, as far as the AC understood, the concern is about the potential robustness of the method. Therefore, this should be addressed by a targeted experiment to examine the robustness of the proposed method.

**Reviewer Scores:**

I think none of the reviewers were positive about this paper. As Reviewer aMaK acknowledged, I also think that this manuscript still requires major revisions.

---

### Decision · Program_Chairs · 2026-01-26

Reject